# A.I.R.: Adaptive, Iterative, and Reasoning-based Frame Selection For Video Question Answering

**Yuanhao Zou**[1][*] **Shengji Jin**[2][*] **Andong Deng**[1]**, Youpeng Zhao**[1]**, Jun Wang**[1]**, Chen Chen**[1][†]
[1]University of Central Florida, [2]Weill Cornell Medicine

## Abstract

Effectively applying Vision-Language Models (VLMs) to Video Question Answering (VideoQA) hinges on selecting a concise yet comprehensive set of frames, as processing entire videos is computationally infeasible. However, current frame selection methods face a critical trade-off: approaches relying on lightweight similarity models, such as CLIP, often fail to capture the nuances of complex queries, resulting in inaccurate similarity scores that cannot reflect the authentic query-frame relevance, which further undermines frame selection. Meanwhile, methods that leverage a VLM for deeper analysis achieve higher accuracy but incur prohibitive computational costs. To address these limitations, we propose *A.I.R.*, a **training-free** approach for **A**daptive, **I**terative, and **R**easoning-based frame selection. We leverage a powerful VLM to perform deep, semantic analysis on complex queries, and this analysis is deployed within a cost-effective iterative loop that processes only a small batch of the most high-potential frames at a time. Extensive experiments on various VideoQA benchmarks demonstrate that our approach outperforms existing frame selection methods, significantly boosts the performance of the foundation VLM, and achieves substantial gains in computational efficiency over other VLM-based techniques. Project Page: https://ucf-air.github.io/

## 1 Introduction

Recent advancements in large Vision-Language Models (VLMs) have revolutionized tasks at the intersection of vision and text (Li et al., 2023; Hurst et al., 2024; Alayrac et al., 2022). Building on their success with static images, the focus has shifted to the more challenging domain of video understanding, which demands a model's ability to reason over complex temporal dynamics and visual narratives. Powerful foundation VLMs are now being adapted to tackle video understanding tasks like Video Question Answering (VideoQA) (Bai et al., 2025; Zhu et al., 2025; Li et al., 2024a).

One critical challenge of video understanding is the perception bottleneck. The extensive number of frames in long videos makes processing the entire video computationally infeasible and exceeds the limited context windows of VLMs. To this end, foundation VLMs typically resort to the frame sampling strategy. The most common one is uniform sampling, which selects frames at a fixed interval. Although uniform sampling maximizes the temporal coverage of the video, its content-agnostic nature usually selects redundant frames while omitting crucial, query-relevant moments.

To alleviate the issue above, a prominent line of research focuses on query-related frame selection. Many of these approaches (Zhang et al., 2025; Sun et al., 2025b; Liu et al., 2025; Tang et al., 2025; Sun et al., 2025a) leverage lightweight multi-modal models like CLIP (Radford et al., 2021) to compute a query-frame similarity, which then guides their proposed algorithm for effective selection. However, to answer a complex query like '*After introducing Tofu making, what kind of traditional technique or scenic spot did the youtuber introduce according to what is shown in the video?*' (Fig. 1 (a)), which requires temporal reasoning (e.g., '*After*') and holistic semantic understanding, a lightweight model lacks these capabilities and instead treats the query as a bag of

---

[*]Equal contribution

[†]corresponding author, chen.chen@crcv.ucf.edu

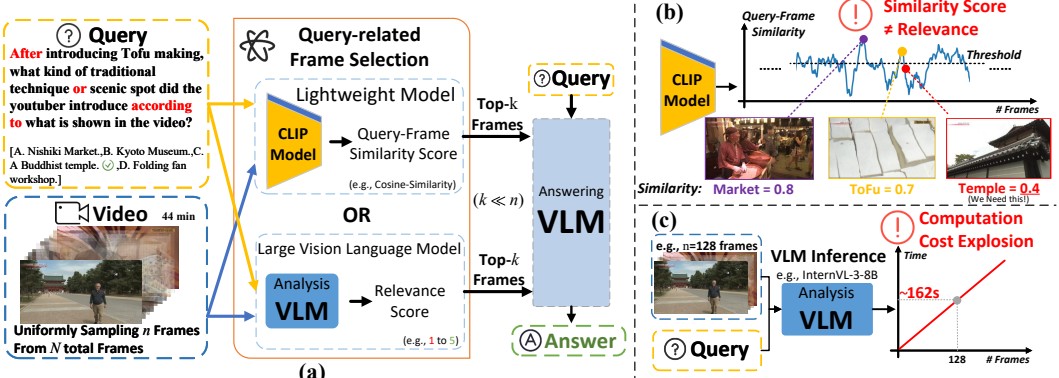

Figure 1: An illustration of the general pipeline of query-related frame selection and the key challenges in query-related frame selection. (a) The pipeline features two branches for query-related frame selection: using a lightweight model (e.g., CLIP) to produce a similarity score between the query and each frame, or using a large Analysis VLM to generate a relevance score. (b) Lightweight models suffer from ambiguous similarity, failing on complex queries. (c) Large VLMs lead to a computation cost explosion scaled with frame number.

keywords Yuksekgonul et al. (2022); Xie et al. (2025). Consequently, as shown in Fig. 1 (b), it incorrectly assigns high similarity scores (score: 0.7) to frame related to the query's context '*Tofu*' or a visually similar frame with '*Market*' (score: 0.8). Crucially, the frame containing the correct option '*A Buddhist Temple*', is ranked lower (score: 0.4) because it has a weaker surface-level association with the query's keywords. Therefore, the lightweight model produces an inaccurate similarity score that fails to reflect the true query-frame relevance, leading to the failure of the selection algorithm.

Taking human perception as an analogy, using a lightweight model (e.g., CLIP) resembles quickly glancing over the video, which may lack sufficient attention and comprehension for complex situations. Other query-related frame selection methods (Hu et al., 2025; Yu et al., 2025; Wang et al., 2025; Ranasinghe et al., 2025; Yu et al., 2023), which employ VLMs as insightful analyzers for each query-frame pair, are more akin to carefully examining each frame. As shown in Fig. 1(a), given a query-frame pair, an analysis VLM generates a relevance score using its strong comprehension of complex queries. An answering VLM is then applied to produce the final answer based on the selected frames. However, this deep analysis comes at a staggering computational cost. As illustrated in Fig. 1 (c), using InternVL-3-8B (Zhu et al., 2025) as an Analysis VLM can take more than one second to process a single frame. For methods that require analyzing an initial set of 128 frames (Hu et al., 2025; Wang et al., 2025), this leads to a prohibitively computational cost explosion (i.e., $\sim 162$ seconds). Therefore, while VLM-based analysis methods are highly effective, their computational cost renders them impractical for many real-world applications.

To address these limitations (poor performance of lightweight models on complex queries and high computational cost of VLM-based analysis), we introduce *A.I.R.*, a **training-free** framework for **A**daptive, **I**terative, and **R**easoning-based frame selection that operates in two stages. In the first stage, it adaptively identifies query-relevant events (i.e., temporal regions) based on the unique distribution of query-frame similarity scores of each video. From these events, it samples an adaptive number of frames per event duration, yielding a high-relevance initial frame set. In the second stage, *A.I.R.* executes an efficient iterative loop that adaptively allocates VLM computation: in each iteration, it ranks all candidates and forwards only a small batch of the high-potential frames for deep, reasoning-based analysis. Frames validated by the VLM then trigger a localized search for additional frames in their temporal neighborhood, uncovering key frames initially assigned with low similarity scores. This synergistic feedback loop, coupled with an Early Stop mechanism, enables *A.I.R.* to converge on the most relevant video segments while requiring only a fraction of the computational cost of conventional VLM-based approaches. Our contributions are threefold:

- We introduce Adaptive Initial Sampling that moves beyond the uniform sampling. It dynamically identifies and samples candidate frames around potential events based on query-frame similarity and controls the output frames via an adaptive budget, robustly handling videos of varied length.

- We propose a novel Iterative Frame Selection algorithm that makes deep VLM analysis computationally tractable, distinguishing itself from prior methods that rely on a computationally expensive, single-pass analysis over large, fixed frame sets.
- Experiments prove that our method can be plug-and-play with diverse foundation VLMs while achieving substantially higher efficiency and accuracy than existing VLM analysis-based methods.

## 2   RELATED WORKS

**Vision Language Models for Video Question Answering.** The paradigm for Video Question Answering (VideoQA) has shifted from early CNN-RNN architectures (Donahue et al., 2015; Venugopalan et al., 2015) to modern Vision-Language Models (VLMs) (Alayrac et al., 2022; Li et al., 2021; Liu et al., 2023b;a). Leveraging the extensive knowledge and reasoning skills acquired during pre-training, these models have spurred two main research directions. The first focuses on models specifically adapted for the video modality, such as Video-LLaVA (Lin et al., 2023) and InternVideo (Wang et al., 2022). A second, more recent trend emphasizes general-purpose VLMs that can process both images and videos, demonstrating strong zero-shot and adaptive capabilities across various tasks. Aligning with this latter direction, our work utilizes a single, powerful VLM—selected from VILA (Lin et al., 2024), QwenVL (Bai et al., 2025; Wang et al., 2024a), LLaVA-OneVision (Li et al., 2024a), and InternVL-3 (Zhu et al., 2025)—to perform both fine-grained analysis of individual frames and high-level video question answering.

**Query-Related Frame Selection via Lightweight Models.** A prominent line of research leverages lightweight models like CLIP (Radford et al., 2021) to efficiently compute query-frame similarity scores. Building on this foundation, various methods have been proposed: BOLT (Liu et al., 2025) applies inverse transform sampling for diversity, MDP3 (Sun et al., 2025b) models the task as a Markov Decision Process, Q-Frame (Zhang et al., 2025) processes frames with dynamic resolutions, and AKS (Tang et al., 2025) introduces an adaptive "Split & Judge" strategy. T* (Ye et al., 2025) applies an iterative temporal search strategy for frame sampling, which first analyzes the objects in the query and then employs an object detector to find frames with relevant grounded objects in each iteration. However, all these methods depend solely on the superficial output scores of lightweight models for their selection logic. In contrast, *A.I.R.* adopts a more sophisticated, two-stage approach. We strategically utilize CLIP only for an initial, coarse-grained sampling, while reserving a powerful VLM for a fine-grained, reasoning-based analysis of complex queries.

**Query-Related Frame Selection via VLMs.** A second branch of methods utilizes powerful VLMs for a deep, semantic analysis of the query-frame relationship. Within this branch, training-free methods like VideoTree (Wang et al., 2025) use online proprietary models (e.g., GPT-4) to generate rich captions for guidance, while others like MVU (Ranasinghe et al., 2025), LLoVi (Zhang et al., 2023), and VideoAgent (Wang et al., 2024b) employ various agents for a comprehensive and reflective analysis. Alternatively, training-based methods such as Frame-Voyager (Yu et al., 2025) fine-tune a VLM to select frames, while SeViLA (Yu et al., 2023) jointly trains a dedicated localizer and an answerer. While effective, these VLM-based methods often perform a computationally expensive, single-pass analysis over a large set of frames. *A.I.R.* overcomes this limitation through a novel iterative loop that makes deep VLM analysis tractable, adaptively allocating computation by analyzing only small, high-potential batches of frames in each step.

## 3   METHOD

### 3.1   OVERVIEW

As illustrated in Fig. 2, our proposed approach, *A.I.R.*, performs frame selection in three stages: Adaptive Initial Sampling, Iterative Frame Selection, and QA Stage. The process begins by sampling $n$ frames from the video (containing $N$ total frames) at a fixed frame rate. As a pre-processing step, these $n$ frames are passed to a CLIP model (Radford et al., 2021) to compute query-frame similarity scores, which is stored as a sparse vector $S \in \mathbb{R}^N$ [1]. This similarity signal $S$ is the input to the Adaptive Initial Sampling stage (Sec. 3.2), which identifies an initial set of $K$ high-potential frame

---

[1] The vector $S \in \mathbb{R}^N$ is sparse as only the $n$ sampled entries have computed values; all other entries with no values are marked with `NaN` and ignored in subsequent operations. Values of $S$ will be updated (Alg. 1).

Figure 2: General pipeline of *A.I.R.* with three stages: (1) **Adaptive Initial Sampling** that identifies potential 'events' based on query similarity and dynamically samples frames around them using an adaptive budget; (2) **Iterative Frame Selection** that progressively refines the frame selection via four steps; and (3) **QA Stage** that feeds the final selected frames into Answering VLM.

indices, $\mathcal{F}_{\text{initial}}$. Subsequently, the Iterative Frame Selection stage (Sec. 3.3, Alg. 1) progressively refines this initial set through a four-step loop. This process yields a final, optimized set of frames, $\mathcal{F}_{\text{final}}^*$. The last one, QA Stage, utilizes an Answering VLM and these selected frames for a one-time inference. Notably, the final number of selected frames, $|\mathcal{F}_{\text{final}}^*| = B$, is not fixed but is determined by an adaptive budget that scales with the video's length (see A.2.1). Finally, we provide a theoretical analysis of our method's efficiency in Sec. 3.4.

## 3.2 ADAPTIVE INITIAL SAMPLING

As shown in Fig. 3 (a), with the initial $n$ frames and the query, unlike other frame selection methods (Liu et al., 2025; Sun et al., 2025b; Tang et al., 2025) that directly work on uniformly sampled frames with their proposed approaches, we perform an Adaptive Initial Sampling. This process selects $K$ query-related frames in advance ($K < n$), which not only provides prior guidance, but also reduces the computation cost for our subsequent Iterative Frame Selection in Sec. 3.3. To achieve this goal, we must first adaptively separate high-relevance frames from low-relevance ones. Hence, we propose an adaptive threshold $T$, inspired by Gaussian Mixture Models (GMMs) (Huang & Chau, 2008; Zhao et al., 2019). Specifically, we hypothesize that the similarity scores $S$ are drawn from a mixture of two underlying distributions: a high-relevance cluster and a low-relevance one. We fit a GMM with two components to model these two clusters. Let the means and standard deviations of them be $(\mu_1, \sigma_1)$ and $(\mu_2, \sigma_2)$. The threshold $T$ is calculated as:

$$T = \max(\mu_1, \mu_2) - \gamma \cdot \max(\sigma_1, \sigma_2),\tag{1}$$

where $\gamma$ is a hyperparameter that controls the stringency of the threshold. This formulation ensures the threshold $T$ is adaptive as it is dynamically computed for each video based on that video's unique distribution of similarity scores. Using the adaptive threshold $T$, we identify an initial set of candidate events $\mathcal{E}'$. An event is defined as any maximal, contiguous temporal region in the video where all similarity scores are at or above the threshold $T$. The events set $\mathcal{E}'$ is formalized as:

$$\mathcal{E}' = \bigcap \left\{ \mathcal{E}'_j = [t_j^{\text{start}}, t_j^{\text{end}}] \mid \forall i \in \mathcal{E}'_j, S_i \geq T \right\},\tag{2}$$

where $\mathcal{E}'_j$ represents the $j$-th event via its start and end frame index. This initial segmentation can be noisy, sometimes splitting a single action into multiple events or creating very short segments. Therefore, we refine $\mathcal{E}'$ with two heuristic-based steps to produce a final set of validated events $\mathcal{E}$: (1) **Merging**: Any two consecutive events separated by a duration less than a minimum distance (i.e., $t_{j+1}^{\text{start}} - t_j^{\text{end}} \leq d_{\min}$) are merged into a single, more coherent event; and (2) **Pruning**: Any event with a total duration less than a minimum length (i.e., $t_j^{\text{end}} - t_j^{\text{start}} \leq l_{\min}$) is then removed.

Finally, with a clean set of refined events $\mathcal{E}$, we perform **Event-Wise Sampling** to select the $K$ initial candidate frames. Our sampling strategy is guided by two key principles, as shown in Fig. 3 (a): (1) Comprehensive coverage: every identified event is represented by at least one frame; and (2) Proportional allocation: longer, more sustained events are allocated a larger portion of the sampling budget. To satisfy these principles, we first calculate the number of frames ($k_j$) to sample from each event based on its relative duration, ensuring that $k_j \geq 1$. We then select $k_j$ peak frames with the highest pre-computed similarity scores from within that event's boundaries. This proportional strategy ensures that the most significant events are more thoroughly represented. The detailed formulation for computing $k_j$ can be found in A.2.2. This process yields an initial sampling set $\mathcal{F}_{\text{initial}} = \{f_1, \ldots, f_K\}$, where $\forall f_i \in \mathcal{F}_{\text{initial}}$ represents frame index and $K < n$. This initial frame set is the input for the subsequent Iterative Frame Selection.

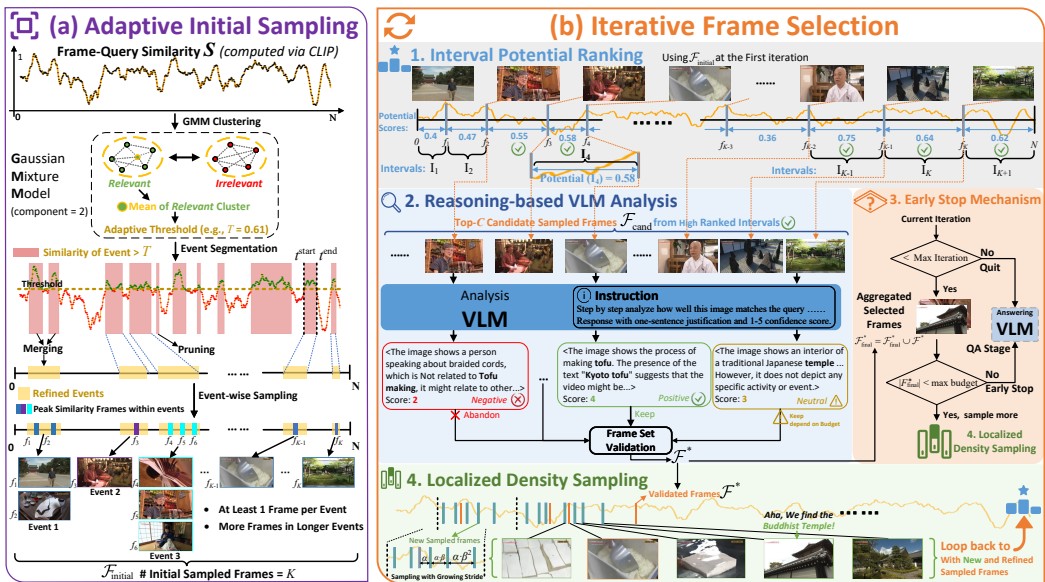

Figure 3: Two main stages in our **A.I.R.**. **(a) Adaptive Initial Sampling**: A GMM-based adaptive threshold is applied to the query-frame similarity $S$ to identify potential events, and then event-wise sampling is conducted on the refined events to obtain $K$ frames ($\mathcal{F}_{\text{initial}}$). **(b) Iterative Frame Selection**: In each iteration, 1) High-potential candidates are selected via Interval Potential Ranking; 2) A VLM performs reasoning-based analysis to validate the best frames; 3) An Early Stop mechanism checks if the frame budget is met; And 4) if not met, the Localized Density Sampling (LDS) discovers more frames around the validated frames and feed them into the next iteration. Notably, LDS is performed on the original video ($N$ frames) instead of the uniformly sampled $n$ frames.

## 3.3 ITERATIVE FRAME SELECTION

While Adaptive Initial Sampling provides a strong set of $K$ candidate frames, analyzing all of them with a powerful VLM remains computationally expensive, as $K$ often scales with the initial sample size $n$. Therefore, we introduce the core of our framework: Iterative Frame Selection. The goal of this stage is to progressively refine the candidate set using an Analysis VLM in a cost-effective manner. This is achieved through a synergistic four-step loop (see Fig. 3 (b) and Alg. 1). In each iteration, **(1) Interval Potential Ranking** first identifies the high-potential candidate frames. **(2)** The candidate frames are then evaluated by the **Reasoning-Based VLM Analysis**, which generates a relevance score and a textual justification for each, allowing us to retain a validated set of truly relevant frames. **(3)** The **Early Stop Mechanism** checks if the adaptive sampling budget (A.2.1) has been met; if so, the process terminates efficiently. **(4)** If the budget is not yet met, **Localized Density Sampling** discovers new, fine-grained frames in the vicinity of the validated frames from (2), which are then fed back into the candidate pool for the next iteration. This cycle of prioritizing, analyzing, and exploring continues until reaching a maximum number of iterations or an early stop is triggered.

**Step 1: Interval Potential Ranking.** Given a set of sampled frames $\mathcal{F}$ (initially, $\mathcal{F} = \mathcal{F}_{\text{initial}}, K = |\mathcal{F}|$) from Adaptive Initial Sampling, the primary role of Interval Potential Ranking is to select a small batch of $C$ high-potential candidates for the Analysis VLM in each iteration. Instead of ranking individual frames by their raw similarity scores, we propose ranking the temporal intervals between these frames. This interval-based approach is more robust as it considers the collective evidence within a temporal region, providing a more comprehensive signal of a potential event than a single frame's score. As illustrated in the first step of Fig. 3 (b), the process begins by partitioning the entire original video into a set of disjoint temporal intervals. These intervals are defined by the indices of the current sampled frames in $\mathcal{F}$. A given interval $I_i$ [2] is formally defined as:

$$I_i = [f_i, f_{i+1}] = \{j \in \mathbb{N} \mid f_i \leq j \leq f_{i+1}\}. \tag{3}$$

---

[2]To ensure the entire video is covered, this partitioning also includes the segments from the start of the video to the first selected frame (i.e., $[1, f_1]$), and from the last selected frame to the video's end (i.e., $[f_K, N]$).

For each interval, we calculate its *potential*—a score indicating its importance to the query—based on its corresponding slice of the pre-computed similarity signal, $S_{f_i:f_{i+1}}$. Inspired by signal processing principles (Boreczky & Rowe, 1996; Wolf, 1996; Liu et al., 2003), this potential is computed as a product of three factors: *Relevance*, *Complexity*, and *Length* (see A.2.4 for details). For a discrete similarity signal $S$, the potential of the interval $\mathrm{I}_i$ is calculated as:

$$\text{Potential}(\mathrm{I}_i) = \underbrace{\text{Mean}(S_{f_i:f_{i+1}})}_{Relevance} \cdot \underbrace{\left(1 + \frac{\sum_{j=f_i}^{f_{i+1}} |S_{j+1} - S_j|}{f_{i+1} - f_i}\right)}_{Complexity} \cdot \underbrace{(1 + c_{\text{len}} \cdot \lg(f_{i+1} - f_i))}_{Length}, \quad (4)$$

where $c_{\text{len}}$ is a hyperparameter that balances the influence of the *Length*. After ranking all intervals by their potential scores, we select the $C$ candidate frames from the highest-ranked intervals (e.g., selected $f_i, f_{i+1}$ from interval $\mathrm{I}_i$) as a candidate set $\mathcal{F}_{\text{cand}}$ for the subsequent VLM analysis.

**Step 2: Reasoning-Based VLM Analysis.** Following the Potential Interval Ranking, the $C$ selected frames $\mathcal{F}_{\text{cand}}$ are analyzed by a Analysis VLM for a focused, reasoning-based evaluation. We leverage the zero-shot, instruction-following capabilities of foundation VLMs to assess the relevance of each frame quantitatively. Guided by a detailed prompt (see Fig. 3 (b) and A.2.5), the VLM is instructed to reason step-by-step, providing both a textual justification and a relevance score (e.g., an integer from 1 to 5) for each candidate frame. Based on the relationship to a predefined threshold $\theta$, these scores are classified as '*Positive*' ($> \theta$), '*Neutral*' ($= \theta$), or '*Negative*' ($< \theta$) and collected into a vector $R \in \mathbb{N}^C$. We retain the '*Positive*' frames to form a validated frame set $\mathcal{F}^*$ as:

$$\mathcal{F}^* = \{f_i \in \mathcal{F}_{\text{cand}} \mid R_i > \theta\}. \quad (5)$$

A fallback mechanism is implemented to handle the case where not enough frames are positively validated (i.e., $|\mathcal{F}^*| < \lfloor B/\mathcal{I}_{\text{max}}\rfloor$, $\mathcal{I}_{\text{max}}$ is the maximum iterations). In this scenario, we instead select $\lfloor B/\mathcal{I}_{\text{max}}\rfloor - |\mathcal{F}^*|$ more frames from the candidate frame set $\mathcal{F}_{\text{cand}}$. The selection is based on a two-tiered priority system: candidates are first grouped by their VLM rating, with frames rated as '*Neutral*' given the highest priority, followed by the '*Negative*' frames. Within each group, the pre-computed similarity score of each frame is then used to determine the final selection order. We then aggregate the validated frames to a cumulative final selection set, $\mathcal{F}_{\text{final}}^*$.

**Step 3: Early Stop Mechanism.** After each round of VLM analysis, we update the cumulative final selection set $\mathcal{F}_{\text{final}}^*$. To maximize efficiency and prevent unnecessary computation, an Early Stop Mechanism is then immediately triggered. We check if the total number of selected frames has met or exceeded the adaptive sampling budget ($|\mathcal{F}_{\text{final}}^*| \geq B$). As illustrated in Fig. 3 (b), if the budget is fulfilled, the iterative loop halts, saving all subsequent VLM analysis costs. If the budget is not yet met, the process continues to the Localized Density Sampling to sample additional frames.

**Step 4: Localized Density Sampling (LDS).** As illustrated in Fig. 3 (b), if the iterative process has not yet met its sampling budget after the Early Stop Mechanism, it proceeds to this final step to discover more candidate frames. Our strategy is motivated by the principle of temporal coherence: the most valuable, undiscovered information is concentrated in the temporal vicinity of the frames VLM has just validated (i.e., $\mathcal{F}^*$). We therefore propose LDS, a search strategy that samples new frames from the original high-frame rate video (from the total $N$ frames instead of $n$), allowing it to capture more fine-grained moments missed by the Adaptive Initial Sampling. LDS employs an exponentially growing sampling stride. This design balances two objectives: it performs a dense, high-resolution search immediately around a validated frame to find precise details, while efficiently exploring the broader context with increasingly sparse samples. For each validated frame $f_i^* \in \mathcal{F}^*$, new sampled frames generated by LDS are formalized as:

$$\text{LDS}(f_i^*) = \left\{\text{round}(f_i^* \pm \alpha \cdot \beta^{(m-1)}) \mid m = 1, 2, \ldots, D\right\}, \quad (6)$$

where $\alpha$ is the initial stride, $\beta > 1$ controls the stride's exponential growth, and $D$ determines the number of frames sampled, which is proportional to the remaining budget, $B - |\mathcal{F}_{\text{final}}^*|$. Crucially, these newly discovered frames are not added directly to the final selection. Instead, they are fed back into the start of the loop. Their query-frame similarity scores are computed to update the signal $S$, and they are added to the candidate pool $\mathcal{F}$ (now $K$ is set to $|\mathcal{F}|$) for the next iteration's Interval Potential Ranking. For instance, as exemplified by the process in Fig. 3 (b), a frame (with '*Tofu*') rated as '*Positive*' in one iteration can trigger a localized search that uncovers the definitive, answer-providing frame (i.e., the frame with '*Buddhist Temple*') via LDS.

## 3.4 EFFICIENCY ANALYSIS

We analyze the core computational efficiency of *A.I.R.* by focusing on the primary bottleneck: the number of frames processed by Analysis VLMs (not Answering VLMs). Conventional VLM-based analysis methods (Hu et al., 2025; Yu et al., 2025; Wang et al., 2025) uniformly sample a large, fixed set of $n_{base}$ frames (e.g., 128 frames) and perform VLM inference on all of them. The total VLM workload for such a method is therefore $n_{base}$ frames. In contrast, *A.I.R.* uses a targeted, iterative strategy. In each iteration, the VLM analyzes only a small batch of $C$ candidate frames. Due to our Early Stop Mechanism, the total number of frames that undergo VLM analysis, $n_{A.I.R.}$, is bounded. In the best-case scenario, the process stops after one iteration, analyzing only $C$ frames, while in the worst-case, it runs for the maximum of $\mathcal{I}_{max}$ iterations. Thus, the VLM workload $n_{A.I.R.}$ is strictly constrained by the best workload $w_{best}$ and the worst workload $w_{worst}$ as:

$$w_{best} \leq n_{A.I.R.} \leq w_{worst}, \tag{7}$$

where $w_{best} = C$ and $w_{worst} = C \cdot \mathcal{I}_{max}$. Our hyperparameter setting (e.g., $C = 12, \mathcal{I}_{max} = 6$) ensures our **worst-case** VLM workload is significantly smaller than conventional methods that analyze a large, fixed number of frames (i.e., $w_{worst} < n_{base} = 128$). Furthermore, *A.I.R.* offers a more intelligent trade-off [3] than methods that achieve efficiency with a small, fixed budget (e.g., $n_{base}$ is 16 or 32, Yu et al. (2023); Ranasinghe et al. (2025); Wang et al. (2024b)). The key advantage of our framework is its adaptivity: the VLM workload only increases as demanded by the video's length (see Tab. 13) and is bounded by $w_{best}$ and $w_{worst}$. This adaptivity allows our method to be more computationally efficient[4] than these fixed-budget approaches (Tab. 7 and Tab. 6).

## 4 EXPERIMENTS

### 4.1 IMPLEMENTATIONS

We use 4 widely-used foundation VLMs as our backbones: VILA-1.5-8B (Lin et al., 2024), QwenVL-2.5-7B (Bai et al., 2025), InternVL-3-8B (Zhu et al., 2025), and LLaVA-OneVision-7B (Li et al., 2024a). We use the same VLM for analysis and answering. Additionally, we employ EVA-CLIP-L (Sun et al., 2023) to compute the query-frame similarity score $S$. We evaluate *A.I.R.* on various long video benchmarks, i.e. Video-MME (Fu et al., 2025), MLVU (Zhou et al., 2025), and LongVideoBench (LVB, Wu et al. (2024)). Besides, we also employ short video benchmarks EgoSchema (Mangalam et al., 2023) and NextQA (Xiao et al., 2021). More details are in A.3. To ensure a fair comparison with other competing frame selection methods, in Tab. 1 and 2, we re-evaluated methods with available code (†) in our controlled environment (*lmms-eval*, Zhang et al. (2024a)), while for others, we compare against their reported metrics (∗) in similar settings.

### 4.2 COMPARISON WITH THE STATE-OF-THE-ART

We conduct a comprehensive evaluation to validate the effectiveness of *A.I.R.* across diverse scenarios, with results presented for long-video benchmarks in Tab. 1 and short-video benchmarks in Tab. 2. To demonstrate the superiority of our approach, we benchmark it against two key categories of methods: foundation VLM baselines, where powerful VLMs use the uniform sampling strategy, and competing state-of-the-art (SoTA) frame selection methods. Our extensive experiments lead to the following key findings: **(1) Powerful Plug-and-Play Enhancement**: As a model-agnostic, training-free module, *A.I.R.*'s benefits generalize across diverse VLMs. For example, on NextQA, applying our method to QwenVL-2.5 results in a massive +7.0 accuracy boost. Similarly, it also provides substantial gains for VILA-1.5, LLaVA-OneVision, and InternVL3, confirming its versatility. **(2) SoTA Accuracy with Superior Efficiency.**: *A.I.R.* elevates foundation VLMs to new SoTA levels while being more frame-efficient. For instance, when paired with InternVL-3 on LVB benchmark, our method achieves a +4.5% absolute gain over the baseline, while analyzing fewer frames on average than the fixed budgets of competitors (i.e., $\leq 32$ vs. 32). **(3) Robust Performance Across Diverse Benchmarks**: While the impact of intelligent frame selection is most critical for

---

[3]We validate this via hyperparameter ablations (i.e., candidate frames $C$ and max iterations $\mathcal{I}_{max}$) in A.4.5.

[4]As shown in Tab. 7, the time cost of other operations, such as initial sampling and frame selection of our method, is minor compared to VLM inference time, so we ignore their theoretical efficiency analysis.

Table 1: Comparison of VLMs and various frame selection methods on Video-MME, MLVU, and LongVideo Bench. ∗ denotes reported results, while † means reproduced ones (see Sec. 4.1).

| Model | LLM Size | #Frames | Video-MME | | MLVU$_{dev}$ | LVB$_{val}$ |
|---|---|---|---|---|---|---|
| | | | w/o sub. | w/ sub. | | |
| *Training-Based Foundation VLMs* | | | | | | |
| LLaVA-OneVision∗ (Li et al., 2024a) | 7B | 32∗ | 58.2 | 61.5 | 64.7 | 56.4 |
| QwenVL-2.5∗ (Bai et al., 2025) | 7B | max: 768 | 65.1 | 71.6 | 70.2 | 45.3 |
| InternVL-3∗ (Zhu et al., 2025) | 8B | max: 64 | 66.3 | 68.9 | 71.4 | 58.8 |
| VILA-1.5∗ (Lin et al., 2024) | 7B | 8 | 47.5 | - | 46.3 | 47.1 |
| *Fair Comparison with Frame Selection Methods* | | | | | | |
| VILA-1.5† | 8B | 8 | 48.9 | 54.2 | 44.7 | 47.9 |
| +Frame-Voyager∗ (Yu et al., 2025) | 8B | 8 | 50.5 | 53.6 | 49.8 | - |
| +MDP3† (Sun et al., 2025b) | 8B | 8 | 53.3 | 57.8 | 52.3 | 52.3 |
| +Q-Frame∗ (Zhang et al., 2025) | 8B | 8 | 50.7 | 55.0 | **54.4** | 51.6 |
| **+Ours** | 8B | 8 | **53.7** | **58.6** | 54.2 | **52.9** |
| QwenVL-2.5† | 7B | 32 | 60.8 | 62.7 | 59.3 | 58.1 |
| +MDP3† | 7B | 32 | 63.8 | 65.7 | 66.2 | 60.0 |
| **+Ours** | 7B | ≤ 32 | **65.0** | **66.3** | **67.5** | **61.4** |
| InternVL-3† | 8B | 32 | 65.6 | 67.3 | 68.4 | 58.3 |
| +MDP3† | 8B | 32 | 66.8 | 69.0 | 74.0 | 60.9 |
| **+Ours** | 8B | ≤ 32 | **68.2** | **69.2** | **74.5** | **62.8** |
| LLaVA-OneVision† | 7B | 32 | 58.5 | 61.7 | 62.4 | 56.6 |
| +AKS∗ (Tang et al., 2025) | 7B | 32 | 58.4 | - | - | 59.3 |
| +MDP3† | 7B | 32 | 60.5 | 64.0 | 68.3 | 59.0 |
| +BOLT∗ (Liu et al., 2025) | 7B | 32 | 59.9 | - | 66.8 | 59.6 |
| **+Ours** | 7B | ≤ 32 | **61.4** | **65.1** | **69.3** | **60.7** |

Table 2: Comparison of VLMs and frame selection methods on Egoschema and NextQA.

| Model | VLM Size | #Frames | Egoschema | | NextQA |
|---|---|---|---|---|---|
| | | | Full | Subset | |
| *VLM analysis-based Frame Selection Methods* | | | | | |
| LLoVi∗ (Zhang et al., 2023) | GPT3.5 | 0.5FPS | 52.2 | - | 66.3 |
| VideoAgent∗ (Wang et al., 2024b) | GPT4 | 1FPS | 54.1 | 60.2 | 71.3 |
| Hu et al. (2025)∗ | 8.5B | 32 | - | 65.9 | 78.4 |
| SeViLA∗ (Yu et al., 2023) | 4.1B | 4 | - | - | 73.8 |
| VideoTree∗ (Wang et al., 2025) | GPT4 | - | 61.1 | 66.2 | 75.6 |
| MVU∗ (Ranasinghe et al., 2025) | 13B | 16 | 37.6 | 60.3 | 55.2 |
| DrVideo (Ma et al., 2025) | GPT4 | - | 61.0 | 66.4 | - |
| T∗ (Ye et al., 2025) | 7B | 8 | - | 66.6 | 80.4 |
| *Fair Comparison with Frame Selection Methods* | | | | | |
| VILA-1.5† | 8B | 8 | 49.5 | 52.8 | 65.9 |
| +Frame-Voyager∗ | 8B | 8 | - | 53.6 | 67.3 |
| +MDP3† | 8B | 8 | 48.5 | 51.0 | 66.1 |
| **+Ours** | 8B | 8 | **50.7** | **53.6** | **70.3** |
| QwenVL-2.5† | 7B | 32 | 57.6 | 59.4 | 74.3 |
| +MDP3† | 7B | 32 | 56.8 | 61.6 | 74.4 |
| **+Ours** | 7B | ≤ 32 | **58.8** | **62.4** | **81.3** |
| InternVL-3† | 8B | 32 | 62.5 | 71.6 | 82.3 |
| +MDP3† | 8B | 32 | 61.6 | 70.0 | 82.3 |
| **+Ours** | 8B | ≤ 32 | **63.3** | **72.2** | **82.6** |
| LLaVA-OneVision† | 7B | 32 | 60.2 | 61.8 | 79.3 |
| +MDP3† | 7B | 32 | 60.3 | 60.8 | 78.9 |
| +BOLT∗ | 7B | 32 | 60.7 | **64.0** | 79.5 |
| **+Ours** | 7B | ≤ 32 | **61.4** | 63.2 | **81.6** |

long videos (Tab. 1), *A.I.R.* also proves effective on shorter video datasets., while it also outperforms competing methods on short-video benchmarks (Tab. 2). More analysis is provided in A.4.1.

## 4.3 ABLATION STUDY

**Influence of Sampled Frame Number.** We investigate how the performance of our approach scales with an increased maximum sampling budget (the upper bound of $B$, see A.2.1). To this end, we adjust the adaptive sampling budget by modifying the maximum frame limit. As illustrated in Fig. 5, our approach consistently outperforms the uniform sampling method on VLMs such as QwenVL-2.5 and LLaVA-OneVision. Furthermore, when compared to CLIP-based query-aware sampling methods like BOLT (Liu et al., 2025), our approach shows an average performance advantage of approximately 1.5%. More detailed ablation study is in Tab. 11 and A.4.3.

**Influence of Different VLM Scales.** As shown in Tab. 3, the results of various scales of VLMs reveal two key findings: (1) *A.I.R.* provides a consistent performance uplift across all tested scales of the QwenVL-2.5 models (7B, 32B, and 72B), confirming the robustness of our method; and (2) We observe that the accuracy gain is most pronounced on the smaller 7B model (+4.2%), while the

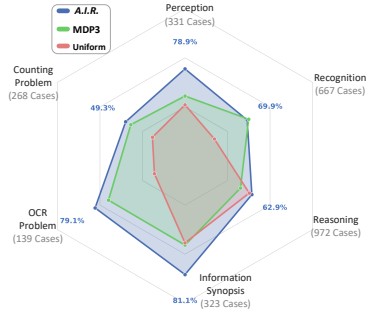

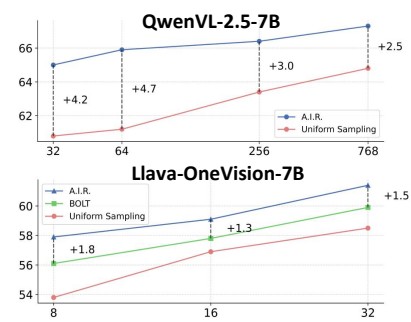

Table 3: Comparison of different VLM scales with **A.I.R.** on Video-MME (w/o subtitle, 32 frames).

| Model | Acc. |
|---|---|
| *QwenVL-2.5-7B*[†] | 60.8 |
| **+Ours** | **65.0** |
| *QwenVL-2.5-32B*[†] | 63.7 |
| **+Ours** | **66.2** |
| *QwenVL-2.5-72B*[†] | 67.0 |
| **+Ours** | **68.2** |
| *GPT-4o*[†] | 61.8 |
| **+Ours** | **65.1** |

Figure 4: Accuracy comparison on 6 question types of Video-MME (w/o sub., 32 frames) using InternVL3-8B.

Figure 5: Accuracy vs. varying sampled frame numbers on Video-MME (w/o subtile) using various foundation VLMs.

Table 4: Ablations of **A.I.R.**'s components on Video-MME using InternVL3-8B. We compare on average frames for answering VLMs and accuracy (Acc.).

| # | Method | Avg. Frames | Acc. |
|---|---|---|---|
| 1 | Uniform Sampling | 32.0 | 65.6 |
| 2 | *A.I.R.* (Our full method) | 24.8 | 68.2 |
| 3 | ↪ w/ Fixed 32 frames (no Ada. Budget) | 32.0 | 68.3 |
| 4 | ↪ w/o Adaptive Similarity Thresholding | 25.5 | 67.3 |
| 5 | ↪ w/o Adaptive Initial Sampling | 25.1 | 66.9 |
| 6 | ↪ w/o Iterative Frame Selection (32 Frames) | 32.0 | 65.2 |
| 7 | ↪ w/o Interval Potential Ranking | 26.2 | 66.7 |
| 8 | ↪ w/o Reasoning-based VLM Analysis | 32.0 | 66.0 |
| 9 | ↪ w/o Localized Density Sampling | 24.5 | 67.2 |

Table 5: Comparison of various CLIP models with **A.I.R.** (using InternVL-3-8B on Video-MME (w/o subtitle) with max sampling budget of 32 frames).

| # | | Acc. |
|---|---|---|
| 1 | Uniform Sampling (32 frames) | 65.6 |
| 2 | *A.I.R.* + CLIP-ViT-B | 66.8 |
| 3 | *A.I.R.* + EVA-CLIP-L (Ours) | **68.2** |
| 4 | *A.I.R.* + LongCLIP-L | 67.4 |
| 5 | *A.I.R.* + CLIP-ViT-L | 67.8 |
| 6 | *A.I.R.* + SigLIP-large | 67.1 |

margin narrows as the base VLM becomes more powerful. This suggests that our frame selection is most critical for smaller models with a weaker intrinsic ability to discern relevance. We further validate the scalability of our plug-and-play framework on state-of-the-art models like GPT-4o, demonstrating consistent performance gains even when scaling to extremely large VLMs.

**Ablation of Components in *A.I.R.* (More Analysis in A.4.4).** We showcase the effectiveness of the individual components of **A.I.R.** in Tab. 4 and Tab. 5. The key findings are: **(1)** Our method significantly outperforms the uniform sampling baseline, while achieving a trade-off between efficiency and accuracy compared to our fixed 32 frames version (Tab. 4). **(2)** Ablating core components, especially the Reasoning-based VLM Analysis, consistently degrades performance, confirming their synergistic contribution (Tab. 4). **(3)** Our framework is robust to the choice of vision encoder, outperforming the baseline across all five CLIP variants that other methods used, and we selected EVA-CLIP-L (Sun et al., 2023) as our default due to its superior performance (Tab. 5).

**Experimental Efficiency Analysis (More Analysis in A.4.5).**

The results in Tab. 6 highlight **A.I.R.**'s superior efficiency-performance trade-off against competing methods on NextQA. When we set the worst workload $w_{worst}$ to 72, which controls the max frames analyzed by VLM, our method not only achieves a SoTA accuracy of 82.6% but does so by adaptively analyzing only 32.2 frames on average. Even with a low worst workload ($w_{worst}$=16), **A.I.R.** still attains an impressive 81.7% accuracy with just 12.4 frames, again outperforming other methods. Tab. 7 further demonstrates **A.I.R.**'s computational efficiency on Video-MME. Tab. 7 illustrates the controlled time comparison of **A.I.R.** against our own baseline, **Direct VLM Analysis**, which mimics a conventional fixed-budget manner that other methods apply using the same VLM (InternVL3-8B). For the intermediate frame analysis (*VLM Analysis Time*), the baseline requires 162.03s to process 128 frames, while **A.I.R.** takes just 42.31s by adaptively processing only 36.5 frames via Iterative Frame Selection. Beyond this *VLM Analysis Time* comparison, the computational overhead of our frame Adaptive Initial Sampling and Iterative Frame Selection stage is minimal, adding up to 0.21s compared to the uniform sampling baseline.

**Generalization to Grounding Task Analysis.** To demonstrate that **A.I.R.**'s adaptive frame selection generalizes beyond VideoQA, we evaluate its performance on Charades-STA (Gao et al., 2017), a challenging temporal grounding benchmark where models must localize specific moments in videos based on natural language queries. As shown in Tab. 8, our training-free method achieves strong performance across all metrics: 59.5% R1@0.3, 39.5% R1@0.5, 18.0% R1@0.7, and 38.8%

Table 6: Comparison of VLM analysis-based frame selection methods on Next-tQA. **#Analyzed** means the number of frames through VLM Analysis. '**TF**' denotes whether training-free or not. We use InternVL3-8B. The same applies to Tab. 7.

| Method | #Analyzed | TF | Acc. |
|---|---|---|---|
| Frame-Voyager (Yu et al., 2025) | 128 | ✗ | 67.3 |
| Hu et al. (2025) | 128 | ✗ | 78.4 |
| VideoTree (Wang et al., 2025) | 128 | ✓ | 75.6 |
| VideoAgent (Wang et al., 2024b) | 48 | ✓ | 71.3 |
| *A.I.R.* ($w_{worst} = 72$) | **32.2** | ✓ | **82.6** |
| SeViLA (Yu et al., 2023) | 32 | ✓ | 73.8 |
| MVU (Ranasinghe et al., 2025) | 16 | ✓ | 55.2 |
| *A.I.R.* ($w_{worst} = 16$) | **12.4** | ✓ | **81.7** |

Table 7: Efficiency comparison on Video-MME. Entries with stage names like '(QA Stage)' denote the time cost for these stages, regardless of any additional VLM analysis time. *VLM Analysis Time* is the time used for analyzing all frames intermediately.

| Method | #Analyzed | Time(s) |
|---|---|---|
| Baseline (Uniform Sampling 32 frames) | - | 0.87 |
| *A.I.R.* (QA Stage) | - | **0.81** |
| *A.I.R.* (Adaptive Initial Sampling) | - | **0.03** |
| *A.I.R.* (Iterative Frame Selection) | - | **0.18** |
| *VLM Analysis Time* | | |
| Direct VLM Analysis | 128 | 162.03 |
| *A.I.R.* ($w_{worst} = 72$) | **36.5** | **42.31** |
| Direct VLM Analysis | 32 | 42.47 |
| *A.I.R.* ($w_{worst} = 32$) | **20.3** | **21.92** |
| Direct VLM Analysis | 16 | 20.39 |
| *A.I.R.* ($w_{worst} = 16$) | **14.1** | **14.61** |

Table 8: Generalization results on temporal grounding benchmark Charades-STA (Gao et al., 2017).

| Model | R1@0.3 | R1@0.5 | R1@0.7 | mIoU |
|---|---|---|---|---|
| *Trained Temporal Grounding VideoLLMs* | | | | |
| VTimeLLM (Huang et al., 2024) | 51.0 | 27.5 | 11.4 | 31.2 |
| HawkEye (Wang et al., 2024c) | 50.6 | 31.4 | 14.5 | 33.7 |
| TimeChat (Ren et al., 2024) | - | 32.2 | 13.4 | 30.6 |
| TimeSuite (Zeng et al., 2024) | **69.9** | **48.7** | **24.0** | - |
| TRACE (Guo et al., 2024) | - | 40.3 | 19.4 | - |
| *General VideoLLMs* | | | | |
| GPT-4o (Hurst et al., 2024) | 55.0 | 32.0 | 11.5 | 35.4 |
| Qwen2.5-VL-7B (Bai et al., 2025) | 44.5 | 30.3 | 15.2 | 30.1 |
| LongVA-7B-DPO (Zhang et al., 2024b) | 22.6 | 10.1 | 2.2 | 14.6 |
| Aria (Li et al., 2024b) | 39.0 | 18.6 | 6.6 | 26.7 |
| *Trained Frame Selection Methods* | | | | |
| GenS (Yao et al., 2025) | 62.9 | 38.7 | 15.2 | 38.0 |
| *Training-Free Frame Selection Methods* | | | | |
| **A.I.R. (Qwen2.5-VL-7B)** | 59.5 | 39.5 | 18.0 | **38.8** |

mIoU. Notably, *A.I.R.* substantially outperforms general-purpose VideoLLMs like Qwen2.5-VL-7B ($44.5\% \rightarrow 59.5\%$ R1@0.3) and even surpasses GPT-4o (32.0% vs. 39.5% R1@0.5). More impressively, our method achieves comparable or superior results to GenS (Yao et al., 2025), a trained frame selection method specifically designed for grounding tasks, while remaining completely training-free. While specialized temporal grounding models like TimeSuite (Zeng et al., 2024) achieve higher performance through task-specific training, our results demonstrate that *A.I.R.*'s adaptive sampling and iterative refinement effectively identify temporally relevant segments without requiring any grounding-specific supervision. This validates the broad applicability of our approach across diverse video understanding tasks.

## 5 CONCLUSIONS

In this paper, we introduced *A.I.R.*, a training-free frame selection approach, which addresses the challenge of efficient and accurate frame selection for Video Question Answering. By employing an Adaptive Initial Sampling stage followed by an iterative, reasoning-driven VLM analysis, *A.I.R.* intelligently focuses computational resources on the most salient temporal regions. Our extensive experiments demonstrate that this approach not only significantly enhances the performance of off-the-shelf VLMs on complex benchmarks but also achieves this with substantially less computational cost than conventional VLM-based analysis methods. By balancing high accuracy with practical efficiency, *A.I.R.* offers a promising path for deploying powerful VLMs in real-world, long-video understanding applications.

## 6 REPRODUCIBILITY STATEMENT

We provide the full details for reproducing our method in the main paper. All datasets used for experiments are publicly available. The partial code of our method, including the code for two core stages (i.e., Adaptive Initial Sampling and Iterative Frame Selection), will be provided in the supplementary materials.

ACKNOWLEDGMENTS

This work used Delta at UIUC NCSA through allocation CIS250367 and 250473 from the Advanced Cyberinfrastructure Coordination Ecosystem: Services & Support (ACCESS) program, which is supported by U.S. NSF grants 2138259, 2138286, 2138307, 2137603, and 2138296.

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

# A APPENDIX

Further details are provided in the appendix. We include our LLM usage statement in A.1, elaborate on our proposed *A.I.R.* framework in A.2, and provide full implementation details in A.3. We present additional quantitative results analysis in A.4, qualitative results analysis in A.5, and limitations in A.6

## A.1 USE OF LLMS

The LLM is an auxiliary tool for our research. Specifically, we use Gemini 2.5 Pro to proofread and polish the paper. No other usages of LLM are precluded from this statement.

## A.2 MORE DETAILS ON OUR METHOD

### A.2.1 ADAPTIVE SAMPLING BUDGET

To ensure our frame selection is efficient for videos of varying lengths, we introduce an adaptive sampling budget $B$ (i.e., the number of frames we need to sample). The budget scales proportionally with the video's duration, allocating fewer frames to short videos (We define the short video as one less than 5 minutes) while respecting the limit of the VLM. Given a video with a total of $N$ frames and a sampling frame rate of FPS, we define the budget $B$ as:

$$B = \max\left(\min\left(\left\lfloor V_{\max} \cdot \frac{n}{300} \right\rfloor, V_{\max}\right), V_{\min}\right), \; n = \frac{\text{FPS} \cdot N}{\text{rate}}, \tag{8}$$

where $n$ is the number of sampled frames, $V_{\max}$ is the maximum allowed number of sampling budget (i.e., max sampling budget), which is constrained by VLM's context length, and $V_{\min}$ is a predefined minimum number of frames required for a robust analysis. $\text{rate}$ is the frame storage rate per second, and for most video files, it is set to $24$. This formulation ensures that the budget scales linearly, while being strictly clamped between the $V_{\min}$ and $V_{\max}$.

### A.2.2 EVENT-WISE SAMPLING

To ensure that longer, more sustained events receive greater attention while guaranteeing all events are represented, we allocate our total sampling budget $K$ proportionally based on the duration of each refined event. Let $L_j = t_j^{\text{end}} - t_j^{\text{start}}$ be the duration of an event $\mathcal{E}_j \in \mathcal{E}$. The number of frames to sample from this event, $k_j$, is calculated using a floor-and-add-one scheme:

$$k_j = \left\lfloor K \cdot \frac{L_j}{\sum_{i=1}^{|\mathcal{E}|} L_i} \right\rfloor + 1, \quad \sum_{j=1}^{|\mathcal{E}|} k_j = K. \tag{9}$$

This formulation ensures both of our key criteria are met, as illustrated in Fig. 3 (a): the proportional term allocates a larger budget to longer events, while the '$+1$' guarantees that even the shortest events are sampled at least once. The set of per-event budgets $k_j$ is then normalized to ensure that the total number of sampled frames is exactly $K$. After determining the budget $k_j$ for each event, we perform the final selection by identifying and choosing the $k_j$ frames that have the highest pre-computed similarity scores from within that event's boundaries. We denote this as the function $\text{Select}(\mathcal{E}_j, \text{Sort}(S_{\mathcal{E}_j}), k_j)$, which means select $k_j$ frames from $\mathcal{E}_j$ based on the sorted $S_{\mathcal{E}_j}$. Finally, we form the initially sampled frame set $\mathcal{F}_{\text{initial}}$ as:

$$\mathcal{F}_{\text{initial}} = \bigcap \left\{ \text{Select}(\mathcal{E}_j, \text{Sort}(S_{\mathcal{E}_j}), k_j) \right\}, \quad \mathcal{E}_j \in \mathcal{E}, \tag{10}$$

where $\bigcap\{\cdot\}$ ensures all selected frames within events are gathered together as an initial frame set $\mathcal{F}_{\text{initial}}$. This frame set will then be used for the Iterative Frame Selection stage.

### A.2.3 ITERATIVE FRAME SELECTION ALGORITHM

We propose an iterative loop with four synergistic steps, as detailed in Alg. 1. In each iteration, the process begins with (1) Interval Potential Ranking. This step identifies temporal intervals between the currently sampled frames, calculates a Potential score $P$ for each interval using Eq. 4, and

---

**Algorithm 1:** Pseudo Code for Iterative Frame Selection

---

**Input:** Analysis VLM, query, sampling budget $B$, query-frame similarity $S$, initial sampled frames
$\quad\quad \mathcal{F}_{\text{initial}}, |\mathcal{F}_{\text{initial}}| = K$, number of candidate frames $C$, max iterations $\mathcal{I}_{\max}$, threshold $T_{\text{positive}}$
**Output:** Finalized Sampled Frames $\mathcal{F}_{\text{final}}^*$.

1   Functions (*Select (F, Sort(S), C): selects C frames from F based on the sorted order of S*)
2   $\mathcal{F} = \mathcal{F}_{\text{initial}}, \mathcal{F}_{\text{final}}^* = \emptyset$;
3   **for** iteration $\leftarrow 1$ *to* $\mathcal{I}_{\max}$ **do**

     /\* `--- 1.   Interval Potential Ranking ---`                     \*/
4      Initialize Intervals $I \in \mathbb{N}^{(K+1)\times 2}$ ;                             /\* `Using Eq.3` \*/
5      Initialize $P \in \mathbb{R}^{K+1}$ for *potential* scores;
6      **for** $i \leftarrow 1$ *to* $K+1$ **do**
7          $P_i = \text{Potential}(I_i)$ ;                                  /\* `Using Eq.4` \*/
8      $\mathcal{F}_{\text{cand}} = \text{Select}(I, \text{Sort}(P), C)$ ;         /\* `Select top-C candidate frames` \*/

     /\* `--- 2.   Reasoning-Based VLM Analysis ---`                  \*/
9      Initialize $R \in \mathbb{N}^C$ for VLM scores;
10     $R = \text{Analysis\_VLM}(query, \mathcal{F}_{\text{cand}})$ ;          /\* `VLM Batch Analysis` \*/
11     $\mathcal{F}^* = \{\mathcal{F}_i \in \mathcal{F}_{\text{cand}} \mid R_i > \theta\}$ ;        /\* `Frame Set refinement using Eq.5` \*/
12     **if** $|\mathcal{F}^*| = 0$ *and* $|\mathcal{F}_{\text{cand}}| > 0$ **then**
13         $\mathcal{F}^* = \text{Select}(\mathcal{F}_{\text{cand}}, \text{Sort}(S), \lfloor B/\mathcal{I}_{\max} \rfloor)$ ;       /\* `Handle edge case` \*/
14     $\mathcal{F}_{\text{final}}^* = \mathcal{F}_{\text{final}}^* \cup \mathcal{F}^*$;

     /\* `--- 3.   Early Stop Mechanism ---`                           \*/
15     **if** $|\mathcal{F}_{\text{final}}^*| \geq B$ **then**
16         $\mathcal{F}_{\text{final}}^* = \text{Select}(\mathcal{F}_{\text{final}}^*, \text{Sort}(S), B)$ ;        /\* `Select top-B frames` \*/
17         Break;

     /\* `--- 4.   Localized Density Sampling (LDS) ---`                \*/
18     **for** $i \leftarrow 1$ *to* $|\mathcal{F}^*|$ **do**
19         $\delta = \text{LDS}(\mathcal{F}_i^*)$ ;                /\* `Sample more frames` ($\delta$) `using Eq.6` \*/
20         $\mathcal{F} = \mathcal{F} \cup \delta$;
21         update $S_\delta$ ;         /\* `Compute and update query-frame similarity` \*/
22     $K = |\mathcal{F}|$;
23   **return** $\mathcal{F}_{\text{final}}^*$

---

selects a small pool of $C$ candidate frames ($\mathcal{F}_{\text{cand}}$) from the highest-potential regions for analysis. These candidates are then evaluated by (2) Reasoning-Based VLM Analysis. The VLM provides a relevance score R for each frame, and only those exceeding a predefined threshold $\theta$ are retained as a set of validated frames, $\mathcal{F}^*$. To ensure robustness, if no frames are validated by the VLM, a fallback mechanism selects a small number based on their initial similarity scores. The validated frames are then added to our final selection pool, $\mathcal{F}_{\text{final}}^*$. The loop is governed by (3) an Early Stop Mechanism, which terminates the process once the total number of selected frames meets the required budget $B$. If the loop continues, the newly validated frames in $\mathcal{F}^*$ serve as anchors for (4) Localized Density Sampling (LDS). For each validated frame, this step generates a new set of fine-grained frames from its immediate temporal vicinity, denoted as $\delta$ in Alg. 1 and elaborated in Eq. 6. These newly discovered frames $\delta$ are then added to the main pool of candidates, enriching the search space for the subsequent iteration. This cycle of prioritizing, validating, and exploring allows our method to efficiently converge on the most critical evidence.

### A.2.4   INTERVAL POTENTIAL RANKING

Inspired by the research in the Signal Processing domain, we define the three factors to rank an interval as: (1) Relevance: The average signal magnitude, analogous to the DC component (Oppenheim, 1999). Higher values suggest greater pertinence; (2) Complexity: The signal's Total Variation, which quantifies volatility and often highlights significant events like scene changes (Rudin et al., 1992); (3) Length: The logarithmic duration of the interval, which prioritizes larger unexplored segments while modeling diminishing returns (Pirolli & Card, 1999). Thus, the *potential* of interval $I_i$

is calculated as:

$$\text{Potential}(\text{I}_i) = \left( \frac{\int_{t=f_i}^{f_{i+1}} S_t \, dt}{f_{i+1} - f_i} \right) \cdot \left( 1 + \frac{\int_{t=f_i}^{f_{i+1}} \left| \frac{d}{dt} S_t \right| dt}{f_{i+1} - f_i} \right) \cdot \lg(f_{i+1} - f_i). \quad (11)$$

Because our similarity signal is discrete, we apply the discrete version (Eq. 4) to compute the potential scores, which serve as a crucial indicator to select high-potential candidates for next-step VLM analysis.

### A.2.5 PROMPTS OF OUR METHOD FOR VLMS

Our method employs a structured, two-stage prompting strategy to guide the Vision Language Models (VLMs) in their respective roles, as illustrated in *Prompt for Analysis VLM* and *Prompt for Answering VLM* below. The first prompt is designed for the Analysis VLM, which functions as a relevance-scoring module. It instructs the VLM to act as an "expert visual reasoner" and evaluate how well each sampled frame matches the user's query. To ensure consistent and quantifiable output, we provide a detailed 5-point scoring rubric, from "1 - Not relevant at all" to "5 - Perfectly matches" (the positive threshold $\theta$ is set to 3). The prompt enforces a strict output format requiring a brief justification sentence and the numerical score, which allows for systematic filtering of visual information. The second prompt is for the Answering VLM, which is responsible for the final decision-making. This prompt is framed as a standard multiple-choice question. It directs the model to synthesize the information from the most relevant frames (identified by the Analysis VLM) and select the correct option. To facilitate straightforward and automated evaluation, the output is constrained to only the letter (A, B, C, or D) of the chosen answer (for dataset with five options, we add one more 'E' here). This dual-prompt design creates a robust pipeline where visual evidence is first critically assessed and filtered before a final answer is determined.

---

**Prompt for Analysis VLM**

You are an expert visual reasoner. Step by step analyze how well this image matches the following query (with options):
`{QUERY}`

Rate the relevance from 1 to 5, using these exact definitions:
1 – Not relevant at all: no relation between image content and the query.
2 – Slightly relevant: only minor contextual hints, but not central to answering.
3 – Moderately relevant: contains preparatory or follow-up context (e.g., setup actions) related to the query.
4 – Highly relevant: shows clear evidence to support the query, but still has ambiguity or missing details.
5 – Perfectly matches: fully sufficient and unambiguous evidence to answer the query.

Avoid overthinking. Trust your immediate judgment.

**Examples:**
- Image of an empty room (no people or objects) → Reasoning: There is nothing related to the query. Score: 1
- Image showing exactly the queried action (clear, direct match) → Reasoning: It perfectly depicts the required event. Score: 5

**Respond only in this format:**

```
Score: <1-5>
Reasoning: <A brief one-sentence justification>
```

---

> **Prompt for Answering VLM**
>
> Select the best answer to the following multiple-choice question based on the video and the subtitles. **Respond with only the letter (A, B, C, or D) of the correct option.**
>
> {QUERY}

### A.3 ADDITIONAL IMPLEMENTATION DETAILS

#### A.3.1 HYPERPARAMETER SETTING

For query-related similarity $S \in \mathbb{R}^N$, we set it as a sparse array with NaN at no-updated entries, so we only focus on the entries with values and skip the computation of the NaN ones. For Adaptive Initial Sampling, we sample frames at 1 FPS (i.e., $\text{FPS} = 1$) and set the minimum frame budget $V_{\min}$ to 8 in Eq. 8, which corresponds to the maximum number of frames accepted by VILA-1.5. For the GMM threshold in Eq. 1, we set $\gamma$ to 0.7. For Event-Wise Sampling, we merge events occurring less than 2 seconds apart (i.e., $d_{\min} = 2 * 24$) and filter out events lasting less than 3 seconds (i.e., $l_{\min} = 3 * 24$), since all video files are stored in 24 fps. We set $K = 2 * \max(B, C)$. In our Iterative Frame Selection (Alg. 1), we set the candidate pool size $C = 12$ (the maximum number accepted to do one batch inference for our machine) and the maximum number of iterations $\mathcal{I}_{\max} = 6$. The VLM rating scores are from 1 to 5 ($\theta = 3$). Finally, for Localized Density Sampling (Eq. 6), we set $\alpha = \max(0.05 \cdot |\text{I}_i|, 15)$, representing 5% of the current interval's length with a minimum value of 10, and set $\beta$ to 1.5.

#### A.3.2 EXPERIMENTAL ENVIRONMENT

All of our experiments are conducted using NVIDIA GH200 GPUs, with Arch64 as the CPU architecture. Our inference is using Pytorch 2.5 with CUDA-11.6 and GCC-11.4.0. One-time experiment of our method is run using 4 aforementioned machines via lmms-eval (Zhang et al., 2024a) inference framework. When using InterVL3 as our foundation VLM, the inference time (Zhu et al., 2025) is around 10 hours. For Video-MME w/o subtitle (Fu et al., 2025). The cost time ranges from 9 hours to 17 hours per different VLMs. Due to the limit of lmms-eval framework, all of our inference is under 1 batch size, while we set the sub-batch size to 12 (e.g., for VLM analysis on frames). For CLIP cache, because each benchmark has queries using the same video repeatedly, we store the image features for quick cosine-similarity computation. All of our videos are loaded from high-res files (all data are provided by *lmms-eval*, https://github.com/EvolvingLMMs-Lab/lmms-eval), so preprocessing operations like *Resizing* may increase the time for computation of CLIP features.

Table 9: Comparison with different methods on Qwen2-VL-7B.

| Model | #Frames | Video-MME | | $\text{LVB}_{val}$ | Egoschema | | NextQA |
| --- | --- | --- | --- | --- | --- | --- | --- |
| | | w/o sub | w/ sub | | Full | Subset | |
| *Qwen2-VL-7B* (Wang et al., 2024a) | 32 | 57.6 | 60.9 | 55.5 | 61.9 | 64.0 | 77.6 |
| +Q-Frame (Zhang et al., 2025) | 32 | 58.3 | 61.8 | 58.4 | - | - | - |
| +AKS (Tang et al., 2025) | 32 | 59.9 | - | **60.5** | - | - | - |
| +Hu et al. (2025) | 32 | 58.7 | - | 57.0 | - | 65.9 | 78.4 |
| **+Ours** | $\leq 32$ | **60.0** | **63.1** | 58.9 | **62.5** | **66.2** | **80.1** |

### A.4 ADDITIONAL RESULTS ANALYSIS

#### A.4.1 ADDITIONAL COMPARISON WITH SoTAs

**Comparison with other SOTA Methods on Qwen2-VL-7B.** Besides the main comparison with SoTAs using QwenVL-2.5, we conducted an additional set of experiments with Qwen2-VL-7B (Wang et al., 2024a) (Tab. 9) to further demonstrate the generalizability and robustness of our *A.I.R.* framework. We compare our performance against a 32-frame uniform sampling baseline as well as several other state-of-the-art (SoTA) frame selection methods. The results are presented in Tab. 9. The data

Table 10: Comparison of Agent-based VLM Frame Selection Methods on Video Question Answering Benchmarks.

| Method | Venue | Base Model | #Frames | Video-MME | | MLVU | LVB | EgoSchema | | NextQA |
|---|---|---|---|---|---|---|---|---|---|---|
| | | | | w/o sub | w/ sub | | | Full | Subset | |
| *Agent-based Frame Selection Methods* | | | | | | | | | | |
| VideoLucy (Zuo et al.) | NeurIPS'25 | DeepSeek-R1 | - | 72.5 | - | 76.1 | - | - | - | - |
| VideoRAG (Luo et al., 2024) | NeurIPS'25 | LLaVA-Video-7B | 64 | - | - | 72.4 | 58.7 | - | - | - |
| T* Ye et al. (2025) | CVPR'25 | LLaVA-OV-7B | 8 | - | - | - | - | - | 66.6 | 80.4 |
| DrVideo (Ma et al., 2025) | CVPR'25 | GPT-4 | - | - | - | - | - | 61.0 | 66.4 | - |
| MemVid (Yuan et al., 2025) | arXiv'25 | Qwen2VL-7B | 128 | 63.7 | 65.7 | - | - | - | - | - |
| VideoAgent2 (Zhi et al., 2025) | arXiv'25 | GPT-4o | - | - | - | - | - | 75.4 | - | 80.5 |
| AKEYS (Fan et al., 2025) | arXiv'25 | GPT-4o | ≤32 | - | - | - | - | 63.6 | 68.6 | 78.1 |
| *Our Frame Selection Method for Comparison* | | | | | | | | | | |
| **A.I.R.** | - | GPT-4o | ≤32 | 65.1 | - | - | - | - | 72.0 | 82.5 |
| **A.I.R.** | - | Qwen2VL-7B | ≤32 | 60.0 | 63.1 | - | 58.9 | 62.5 | 66.2 | 80.1 |
| **A.I.R.** | - | InternVL3-8B | ≤32 | 68.2 | 69.2 | 74.5 | 62.8 | 63.3 | 72.2 | 82.6 |
| **A.I.R.** | - | LLaVA-OV-7B | ≤32 | 61.4 | 65.1 | 69.3 | 60.7 | 61.4 | 63.2 | 81.6 |

shows that **A.I.R.** consistently outperforms both the baseline and competing methods on this new VLM backbone. Notably, on the Video-MME (w/o sub) benchmark, our approach achieves the highest accuracy of 60.0%, surpassing strong methods like AKS (59.9%) and Q-Frame (58.3%), and providing a significant +2.4% uplift over the baseline. Our method also achieves the top performance on Egoschema and NextQA. While AKS shows a strong result on LVB, our method remains highly competitive with a score of 58.9%, a substantial improvement over the baseline's 55.5%. This experiment validates that the benefits of our adaptive, iterative, and reasoning-based approach are not tied to a specific VLM architecture and can be effectively generalized.

**Comparison with Agent-based Frame Selection Methods.** To further validate A.I.R.'s effectiveness against cutting-edge approaches, we compare against seven recent agent-based methods from NeurIPS'25, CVPR'25, and arXiv'25 in Tab. 10. These methods leverage multi-step reasoning and planning capabilities of large language models, often using frontier models like GPT-4o or DeepSeek-R1. Our results reveal three key findings: **(1) Superior performance with matched backbones and exceptional frame efficiency.** When using identical base models, A.I.R. consistently outperforms agent-based methods while maintaining high frame efficiency. For instance, with GPT-4o and comparable frame budgets (≤32 frames), A.I.R. surpasses AKEYS on EgoSchema Subset by +3.4% and NextQA by +4.4%. Similarly, with LLaVA-OneVision-7B, A.I.R. exceeds T* on NextQA (+1.2%). Moreover, compared to VideoRAG which uses 64 frames with LLaVA-Video-7B to achieve 58.7% on LVB, A.I.R. with LLaVA-OneVision-7B reaches 60.7% with only ≤32 frames, achieving 2× frame efficiency with +2.0% accuracy improvement. **(2) Competitive performance with smaller models.** Remarkably, A.I.R. with the 8B-scale InternVL3 achieves performance competitive with or superior to agent-based methods using much larger frontier models. On NextQA, InternVL3-8B with A.I.R. reaches 82.6%, surpassing VideoAgent2's GPT-4o result (80.5%) and AKEYS's GPT-4o result (78.1%). On EgoSchema, InternVL3-8B achieves 72.2% (Subset) and 63.3% (Full), exceeding DrVideo's GPT-4 performance (66.4% Subset, 61.0% Full). This demonstrates that A.I.R.'s intelligent frame selection can unlock the full potential of smaller, open-source models to match or exceed the performance of expensive proprietary models. **(3) Comprehensive and transparent evaluation.** Unlike many agent-based methods that report results selectively across 2-3 benchmarks, A.I.R. provides consistent results across all five major benchmarks (Video-MME, MLVU, LVB, EgoSchema, NextQA) for multiple VLM backbones, demonstrating robustness and eliminating concerns about cherry-picked results. While agent-based methods offer powerful multi-step reasoning capabilities, they face practical challenges including high computational costs from multiple LLM calls and dependency on proprietary models. In contrast, A.I.R. achieves competitive performance through a streamlined, training-free framework that seamlessly integrates with any VLM backbone.

### A.4.2 DETAILED ANALYSIS OF QUESTION TYPE

As illustrated in Fig. 4, we evaluate the performance of our proposed method, A.I.R., against two baseline approaches: MDP3 (Sun et al., 2025b) and Uniform Sampling, across six distinct problem domains. The results clearly demonstrate that **A.I.R.** consistently and substantially outperforms both baselines across all evaluated categories. Notably, **A.I.R.** exhibits exceptional proficiency in tasks requiring textual and holistic understanding, achieving its highest scores in Information Synopsis (81.1%), OCR Problems (79.1%), and Perception (78.9%). It also maintains a strong lead in Recog-

Table 11: Ablation study on different frame numbers for training-free methods.

| Model | #Frames | Video-MME | | MLVU$_{dev}$ | LVB$_{val}$ | Egoschema | | NextQA |
|-------|---------|-----------|--|--------------|-------------|-----------|--|--------|
| | | w/o sub | w/ sub | | | Full | Subset | |
| *QwenVL-2.5* | 32 | 60.8 | 62.7 | 59.3 | 58.1 | 57.6 | 59.4 | 74.3 |
| **+Ours** | ≤ 32 | **65.0** | **66.3** | **67.5** | **61.4** | **58.8** | **62.4** | **81.3** |
| *QwenVL-2.5* | 64 | 61.2 | 65.0 | 63.8 | 59.0 | 58.8 | 62.0 | 75.9 |
| **+Ours** | ≤ 64 | **65.9** | **67.2** | **69.7** | **62.5** | **59.8** | **63.2** | **82.5** |
| *QwenVL-2.5* | 256 | 63.4 | 67.3 | 67.8 | 60.2 | 58.8 | 62.6 | 74.3 |
| **+Ours** | ≤ 256 | **66.4** | **67.9** | **71.7** | **62.8** | **59.9** | **63.6** | **83.3** |
| *LLaVA-OneVision* | 8 | 53.8 | 58.9 | 58.9 | 54.2 | 59.2 | 62.0 | 77.4 |
| +BOLT* | 8 | 56.1 | - | 63.4 | 55.6 | 59.2 | 62.2 | 77.4 |
| **+Ours** | 8 | **57.9** | **60.5** | **63.8** | **57.3** | **59.4** | **63.4** | **78.9** |
| *LLaVA-OneVision* | 16 | 56.9 | 60.0 | 61.2 | 55.7 | 59.5 | 61.4 | 78.1 |
| +BOLT* | 16 | 57.8 | - | **65.8** | 57.0 | 59.9 | 61.8 | 78.3 |
| **+Ours** | ≤ 16 | **59.1** | **60.8** | 65.6 | **58.9** | **60.4** | **62.2** | **79.4** |
| *LLaVA-OneVision* | 32 | 58.5 | 61.7 | 62.4 | 56.6 | 60.2 | 61.8 | 79.3 |
| +BOLT* | 32 | 59.9 | - | 66.8 | 59.6 | 60.7 | 64.0 | 79.5 |
| **+Ours** | ≤ 32 | **61.4** | **65.1** | **69.3** | **60.7** | **61.4** | **64.2** | **81.6** |

nition (69.9%) and Reasoning (62.9%), showcasing its robust capabilities on a high volume of test cases. Interestingly, the most challenging domain for all methods, including *A.I.R.*, is the Counting Problem, where *A.I.R.* scores 49.3%. While this is its lowest score, it still represents a significant margin of improvement over both MDP3 and Uniform Sampling. In comparison, the MDP3 method offers a moderate improvement over the basic Uniform Sampling baseline but is clearly surpassed by the advanced capabilities of *A.I.R.* in every task. Overall, the analysis highlights the comprehensive strengths and superior performance of the *A.I.R.* model across a diverse set of complex problems.

### A.4.3 Additional Ablations of Sampled Frame Number

In this section, we provide a detailed ablation study on the effect of the maximum sampling budget on the performance of our *A.I.R.* framework. We compare our method against a uniform sampling baseline and other state-of-the-art methods across various benchmarks, using two distinct VLM backbones: QwenVL-2.5 and LLaVA-OneVision. The comprehensive results are presented in Tab. 11.

The analysis on the QwenVL-2.5 backbone demonstrates that *A.I.R.* provides a substantial and consistent performance improvement over the uniform sampling baseline at every tested budget (32, 64, and 256 frames). For instance, with a maximum of 32 frames, our method boosts performance on Video-MME (w/o sub) from 60.8% to 65.0% and on MLVU from 59.3% to 67.5%. As the maximum budget increases, the performance of our method continues to scale effectively, consistently maintaining a significant advantage over the baseline, which sees only modest gains from having more frames.

The three-way comparison on the LLaVA-OneVision backbone further validates our approach. Across nearly all settings, *A.I.R.* outperforms both the uniform sampling baseline and the strong competitor, BOLT. For example, with a 32-frame budget, *A.I.R.* achieves 61.4% on Video-MME, surpassing both the baseline (58.5%) and BOLT (59.9%), and scores 81.6% on NextQA, again outperforming both the baseline (79.3%) and BOLT (79.5%). The effectiveness of our intelligent selection is highlighted by the fact that *A.I.R.* with a maximum of 16 frames (59.1% on Video-MME) often outperforms the uniform sampling baseline that uses a fixed 32 frames (58.5%). This comprehensive analysis of two different VLMs confirms that our adaptive and iterative framework is a robust and highly effective strategy for improving video understanding across a wide range of frame budgets.

### A.4.4 Additional Analysis of Method Ablations

**Influence of Method Components.** To validate the contribution of each component within the *A.I.R.* framework, we conduct a systematic ablation study, with results presented in Tab. 4. Our full method (#2) achieves 68.2% accuracy, a significant improvement of +2.6% over the Uniform

Table 12: Ablation study on the hyperparameters $C$ (candidate pool size) and $\mathcal{I}_{\max}$ (max iterations). The trade-off between efficiency (Time) and performance (Accuracy) is evaluated on NextQA using InternVL3-8B. The chosen default configuration is highlighted. Times (in seconds) represent the VLM inference time, while (16.0f) represents the overall frames analyzed by VLM.

| # | Max Sampling Budget ($V_{\max}$) | Candidate Pool Size ($C$) | Max Iterations ($\mathcal{I}_{\max}$) | Time (s) | Accuracy (%) |
|---|---|---|---|---|---|
| 1 | 16 | 8 | 2 | 14.87 (13.6f) | 81.61 |
| 2 | 16 | 4 | 4 | 12.62 (12.4f) | 81.68 |
| 3 | 16 | 8 | 4 | 18.92 (20.3f) | 81.92 |
| 4 | 32 | 8 | 4 | 29.17 (27.2f) | 82.20 |
| 5 | 32 | 12 | 6 | 34.24 (32.2f) | 82.63 |
| 6 | 32 | 16 | 8 | 36.52 (35.6f) | **82.91** |

Sampling baseline (#1), while also using fewer frames on average. Disabling individual components generally degrades performance, confirming their synergistic importance. The most critical component is the entire Iterative Frame Selection stage (#6); its removal causes the largest accuracy drop to 65.2%, demonstrating that the iterative refinement process is essential for identifying relevant frames. The Reasoning-based VLM Analysis (#8) is also highly impactful, with its removal reducing accuracy to 66.0% and increasing the frame count to 32. The Interval Potential Ranking (#7) and Adaptive Initial Sampling (#5) are similarly important, with their removal reducing accuracy to 66.7% and 66.9%, respectively. Notably, replacing our adaptive budget with a fixed 32-frame budget (#3) yields a marginal 0.1% increase in accuracy to 68.3%. However, we selected our adaptive approach as the default because it strikes a superior efficiency-performance trade-off: it achieves nearly identical top-tier performance while using 22.5% fewer frames on average (24.8 vs. 32.0), making it a more practical and scalable solution.

**Ablation on CLIP Models.** We analyze the impact of different CLIP variants on the *A.I.R.* framework in Tab. 5. The results show that all configurations utilizing our method demonstrate a marked improvement over the Uniform Sampling baseline, highlighting the fundamental effectiveness of our framework irrespective of the specific vision backbone employed. Among the tested variants, the combination of *A.I.R.* with EVA-CLIP-L achieves the highest performance, reaching an accuracy of 68.2%. We also observe that larger models generally provide better results, with CLIP-ViT-L outperforming its base-sized counterpart by 1.0%. Based on this analysis, we selected EVA-CLIP-L as the default vision encoder for our main experiments due to its superior empirical performance.

### A.4.5 Additional Efficiency and Hyperparameter Analysis

We analyze the impact of our key hyperparameters—Max Sampling Budget ($V_{\max}$), Candidate Pool Size ($C$), and Max Iterations ($\mathcal{I}_{\max}$)—on the trade-off between performance and efficiency, with results on NextQA presented in Tab. 12. The results demonstrate that various *A.I.R.* configurations can achieve state-of-the-art accuracy while remaining highly frame-efficient. For instance, our settings with a 16-frame budget (rows #1-#3) surpass 81.6% accuracy using fewer than 14 frames on average, significantly outperforming fixed-budget baselines that score much lower with a similar frame count. Focusing on the configurations with $V_{\max} = 32$ (rows #4-#6) allows for a deeper analysis of this trade-off. The setting in row #4 ($C = 8, \mathcal{I}_{\max} = 4$) establishes a strong baseline at 82.20% accuracy with 27.2 analyzed frames. Our chosen default setting (row #5, $C = 12, \mathcal{I}_{\max} = 6$) improves accuracy to 82.63% for a modest increase in VLM workload to 32.2 frames. Further increasing the hyperparameters (row #6) yields the highest accuracy (82.91%), but this marginal +0.28% gain requires a larger workload (35.6 frames), indicating a clear point of diminishing returns. We therefore select the configuration from row #5 as our default, as it strikes an optimal balance between high performance and computational cost. Overall, these results show that *A.I.R.* is robust to hyperparameter changes and our chosen default provides the best efficiency-performance trade-off.

In Tab. 13, we provide a granular efficiency analysis by breaking down the performance of *A.I.R.* across short, medium, and long videos. This detailed view reveals several key advantages of our adaptive framework:

- Adaptive Cost vs. Fixed Waste. The results show that A.I.R.'s computational cost intelligently scales with video length, in stark contrast to conventional fixed-budget methods. For instance, a 128-frame baseline incurs a constant, high cost of 162.03s regardless of video duration. A.I.R.,

Table 13: Detailed Efficiency Comparison on Video-MME using InternVL3-8B.

| Model | Short Video $\leq 5min$ | Medium Video $5 - 15min$ | Long Video $\geq 15min$ | Overall *Avg.* 17 min |
|---|---|---|---|---|
| CLIP Inference (1FPS) | 21.30 | 40.32 | 78.55 | 46.72 |
| Uniform Sampling (Answering) | 0.56 | 0.98 | 1.06 | 0.87 |
| *A.I.R.* (Sampling, Selection & Answering) | **0.63** | **1.10** | **1.32** | **1.02** |
| Conventional VLM inference | 162.03 (128f) | 162.03 (128f) | 162.03 (128f) | 162.03 (128f) |
| *A.I.R.* (VLM inference, $V_{\max} = 32, C = 12, \mathcal{I}_{\max} = 6$) | **26.34** (25.3f) | **46.17** (37.4f) | **54.42** (46.7f) | **42.31** (36.5f) |
| Conventional VLM inference | 42.47 (32.0f) | 42.47 (32.0f) | 42.47 (32.0f) | 42.47 (32.0f) |
| *A.I.R.* (VLM inference, $V_{\max} = 16, C = 8, \mathcal{I}_{\max} = 4$) | **14.45** (12.5f) | **18.90** (17.4f) | **32.41** (30.9f) | **21.92** (20.3f) |
| Conventional VLM inference | 20.39 (16.0f) | 20.39 (16.0f) | 20.39 (16.0f) | 20.39 (16.0f) |
| *A.I.R.* (VLM inference, $V_{\max} = 16, C = 4, \mathcal{I}_{\max} = 4$) | **9.05** (9.7f) | **13.47** (12.8f) | **21.31** (19.8f) | **14.61** (14.1f) |

however, is far more efficient on short videos (e.g., 26.34s in our default setting) and, while its cost increases for longer videos (54.42s), it remains nearly three times faster than the fixed-budget approach, effectively eliminating wasteful computation.

- Intelligent, Sub-Linear Scaling. Crucially, the VLM workload for *A.I.R.* does not scale linearly with the video's duration. In our default setting ($V_{\max} = 32$), the number of analyzed frames grows from 25.3 for short videos to 46.7 for long videos. This sub-linear growth suggests that *A.I.R.* is effective at pinpointing information-dense regions, tying its computational cost more closely to the query's informational complexity rather than the video's raw length.

- Minimal Framework Overhead. The top section of the table highlights the efficiency of the core algorithm itself. The total pipeline time for *A.I.R.* (1.02s on average) is remarkably close to the fastest Uniform Sampling baseline (0.87s). This demonstrates that the complex logic of adaptive sampling, ranking, and iteration adds minimal computational overhead, making the entire framework lightweight and practical.

Taken together, this granular analysis confirms that *A.I.R.* is a scalable solution that intelligently manages its VLM workload based on the specific characteristics of the video, making it highly practical for real-world scenarios with varying video lengths.

### A.4.6 COMPREHENSIVE HYPERPARAMETER ABLATIONS

To validate the robustness of *A.I.R.* across different hyperparameter configurations, we conduct a systematic ablation study on LongVideoBench (LVB) and NextQA using InternVL3-8B. As shown in Tab. 14, we independently vary each of the five key hyperparameters while keeping others fixed at their default values: $\gamma = 0.7$ (GMM coefficient), $\alpha = 15$ (minimum frame distance), $\beta = 1.5$ (LDS growth factor), $l_{\min} = 20$ (minimum event length for pruning), and $d_{\min} = 2.0s$ (minimum gap for merging events).

**Individual Parameter Analysis.** Our results demonstrate strong robustness across all hyperparameters. For the GMM coefficient $\gamma$ (rows 1-3), we observe that moderate values (0.5-0.7) yield the best performance, with $\gamma = 0.5$ and $\gamma = 0.7$ both achieving 82.6% on NextQA. Extreme values ($\gamma = 0.3$ or $\gamma = 0.9$) lead to slightly degraded performance (-0.3% to -0.4%), as they either select too many low-relevance frames or miss important events. For minimum frame distance $\alpha$ (rows 4-6), larger values (20-25) prevent redundant sampling but may miss fine-grained details, while smaller values (10) enable denser coverage. The LDS growth factor $\beta$ (rows 7-9) shows consistent performance across the tested range (1.2-2.0), with moderate values (1.5-1.7) performing slightly better. The pruning threshold $l_{\min}$ (rows 10-12) and merging gap $d_{\min}$ (rows 13-14) both demonstrate stability, with performance varying by less than 1.0% across different settings.

**Strategy-Based Combinations.** We further test four strategy combinations (rows 15-18) designed for different video characteristics: (15) Short-Video strategy with dense sampling ($\gamma = 0.5, \alpha = 10, l_{\min} = 15, d_{\min} = 1$); (16) Long-Video strategy with sparse sampling ($\alpha = 25, l_{\min} = 30, d_{\min} = 3$); (17) Conservative strategy with strict thresholds ($\gamma = 0.3, \alpha = 20$); and (18) Aggressive strategy with loose thresholds ($\gamma = 0.9, \alpha = 20$). All combinations maintain competitive performance (61.7-62.9% on LVB, 81.8-82.7% on NextQA), demonstrating that *A.I.R.* can be flexibly adapted to different video types while preserving effectiveness.

Table 14: Systematic ablation study of *A.I.R.* hyperparameters using InternVL3-8B. Default configuration (gray) uses parameters from Appendix A.3.1: $\gamma = 0.7$, $\alpha = 15$, $\beta = 1.5$, $l_{\min}$=20, $d_{\min}$=2.0s. We vary each parameter independently to isolate effects. All configurations maintain performance within ±1.1% of baseline, demonstrating strong robustness. Rows 15-18 test strategy combinations: Short-Video (dense sampling), Long-Video (sparse sampling), Conservative (strict), Aggressive (loose). All use $V_{\max} = 32$, $C = 12$, $\mathcal{I}_{\max} = 6$.

| # | $\gamma$ | $\alpha$ | $\beta$ | $l_{\min}$ | $d_{\min}$ | LVB | NextQA |
|---|---|---|---|---|---|---|---|
| - | **0.7** | **15** | **1.5** | **20** | **2** | **62.8** | **82.6** |
| *$\gamma$ ablation (GMM coefficient)* | | | | | | | |
| 1 | 0.3 | 15 | 1.5 | 20 | 2 | 62.2 | 82.3 |
| 2 | 0.5 | 15 | 1.5 | 20 | 2 | 62.6 | 82.6 |
| 3 | 0.9 | 15 | 1.5 | 20 | 2 | 62.0 | 82.2 |
| *$\alpha$ ablation (min_frame_distance)* | | | | | | | |
| 4 | 0.7 | 10 | 1.5 | 20 | 2 | 62.4 | 82.8 |
| 5 | 0.7 | 20 | 1.5 | 20 | 2 | 62.8 | 82.1 |
| 6 | 0.7 | 25 | 1.5 | 20 | 2 | 62.9 | 82.0 |
| *$\beta$ ablation (LDS growth factor)* | | | | | | | |
| 7 | 0.7 | 15 | 1.2 | 20 | 2 | 62.4 | 82.4 |
| 8 | 0.7 | 15 | 1.7 | 20 | 2 | 62.6 | 82.5 |
| 9 | 0.7 | 15 | 2.0 | 20 | 2 | 62.1 | 82.1 |
| *$l_{\min}$ ablation (Pruning)* | | | | | | | |
| 10 | 0.7 | 15 | 1.5 | 15 | 2 | 62.0 | 82.4 |
| 11 | 0.7 | 15 | 1.5 | 25 | 2 | 62.8 | 82.1 |
| 12 | 0.7 | 15 | 1.5 | 30 | 2 | 62.7 | 81.8 |
| *$d_{\min}$ ablation (Merging)* | | | | | | | |
| 13 | 0.7 | 15 | 1.5 | 20 | 1 | 62.3 | 82.8 |
| 14 | 0.7 | 15 | 1.5 | 20 | 3 | 62.6 | 82.3 |
| *Strategy-based combinations* | | | | | | | |
| 15 | 0.5 | 10 | 1.5 | 15 | 1 | 61.7 | 82.7 |
| 16 | 0.7 | 25 | 1.5 | 30 | 3 | 62.9 | 81.8 |
| 17 | 0.3 | 20 | 1.7 | 20 | 2 | 61.8 | 82.0 |
| 18 | 0.9 | 20 | 1.2 | 20 | 2 | 62.1 | 81.9 |

**Robustness Analysis.** Critically, all 18 configurations maintain performance within ±1.1% of the default setting on both benchmarks, confirming that *A.I.R.* is highly robust to hyperparameter choices.

## A.5 QUALITATIVE RESTULTS ANALYSIS

As shown in Fig. 6 and Fig. 7, *A.I.R.*'s strength lies in its iterative refinement process. The system initially casts a wide net and then uses a Vision-Language Model (VLM) to progressively score and filter video frames. For instance, when searching for a character's post-meeting actions, the VLM intelligently discards irrelevant scenes like pre-meeting preparations. A localized sampling mechanism then focuses on temporal regions around these high-scoring frames, creating a feedback loop that efficiently uncovers more relevant content. This dynamic process allows *A.I.R.* to converge on the precise moments that answer a query—whether it's a character having lunch or an artist applying glaze—yielding a final selection of frames that is both comprehensive and highly relevant.

*A.I.R.*'s superiority is further demonstrated in direct comparisons against baseline methods in Fig. 8 and Fig. 9. Baselines like Uniform Sampling are indiscriminate, capturing disconnected or irrelevant moments (e.g., logos, underwater scenes) due to their fixed-interval nature. While an improvement, CLIP Top-K often falls into "relevance traps," leading to high redundancy by over-sampling visually similar frames (like a car scene) or wasting its budget on tangentially related content (people near a volcano instead of the lava flow). In stark contrast, *A.I.R.* constructs a coherent narrative. It precisely captures the key chronological events—from establishing shots to the core lava formation, or a Vlogger's complete daily routine—while ensuring temporal diversity. These results confirm

that our adaptive approach overcomes the critical limitations of simpler methods, achieving superior semantic relevance and temporal coherence.

## A.6 LIMITATIONS

We acknowledge several limitations of our current framework. First, the performance of *A.I.R.* is fundamentally bound by the capabilities of its underlying analysis VLM. For instance, as shown across our benchmarks (Tab. 1 and 2), using a more advanced model like InternVL-3-8B yields significantly better results than less capable models such as VILA-1.5, even at a comparable parameter size. Additionally, a known shortcoming of most frame selection methods, including ours, is their performance on fine-grained tasks like object counting, as highlighted in Fig. 4. Moreover, our approach currently processes only the visual track, omitting crucial information that may be present in the audio. Finally, while significantly more efficient than brute-force analysis, the iterative nature of our framework introduces a computational latency that may not be suitable for real-time applications.

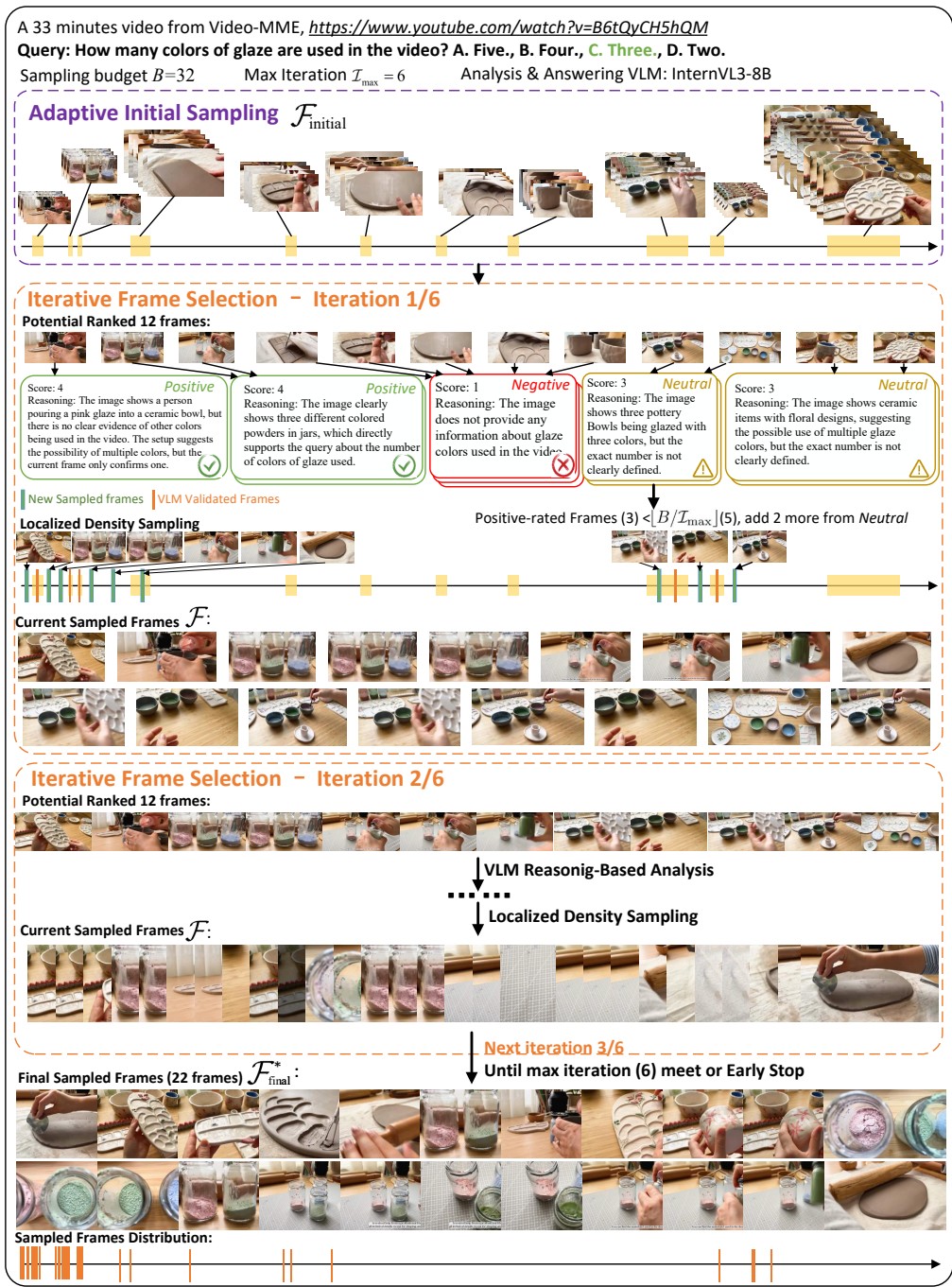

Figure 6: Detailed workflow of *A.I.R.* on a 33-minute Video-MME example showing the iterative frame selection process for the query "*How many colors of glaze are used in the video?*" with VLM reasoning scores and localized density sampling.

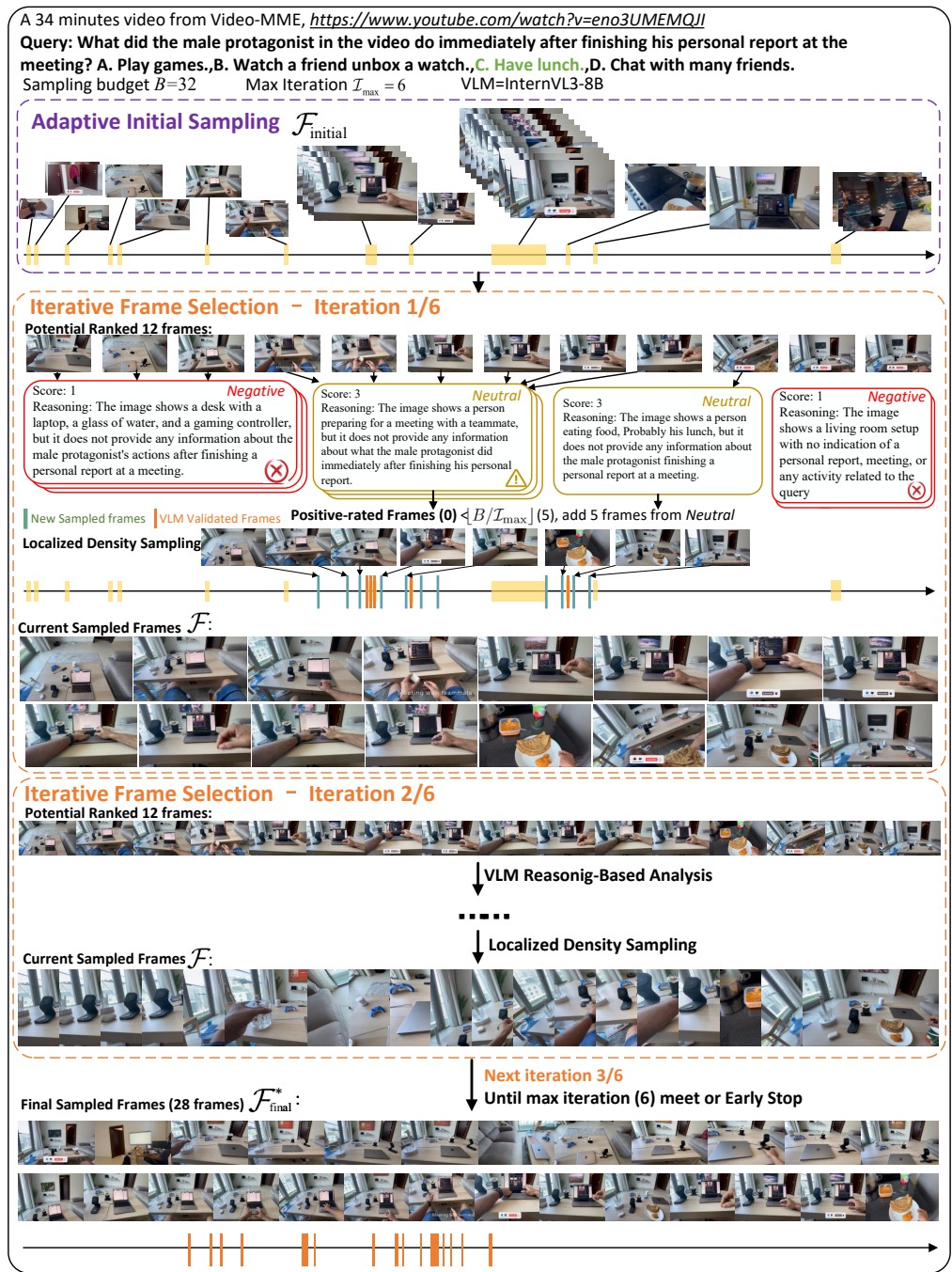

Figure 7: Detailed workflow of **A.I.R.** on a 34-minute Video-MME example showing the iterative frame selection process for the query "*What did the male protagonist in the video do immediately after finishing his personal report at the meeting?*" with VLM reasoning scores and localized density sampling.

Question: How was Nahuku formed according to the video?
A. It was created by lava tubes.      B. It was created by a river of molten lava.
C. It was formed by the workers in the park.      D. It was formed due to the function of wind.

## Uniform Sampling

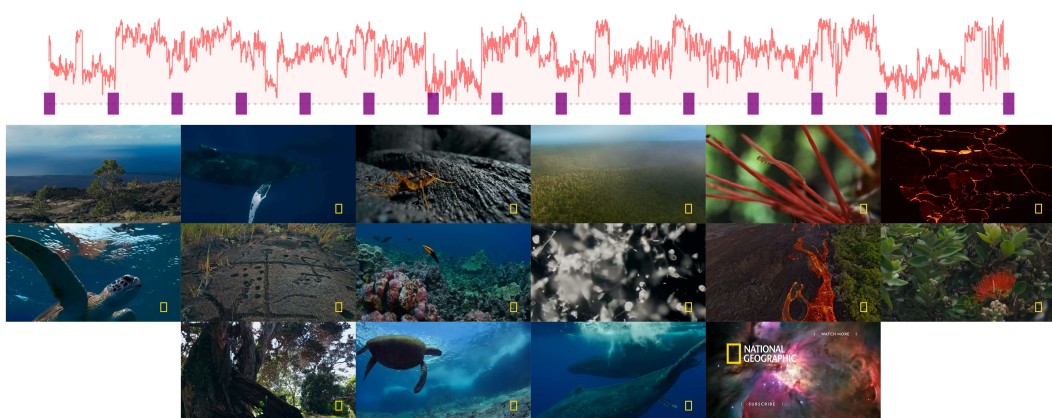

## CLIP (Top-K)

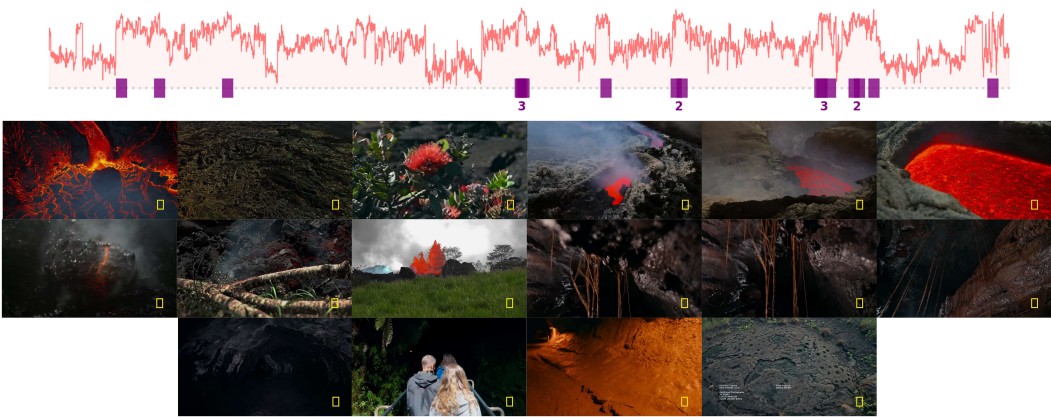

## *A.I.R.*

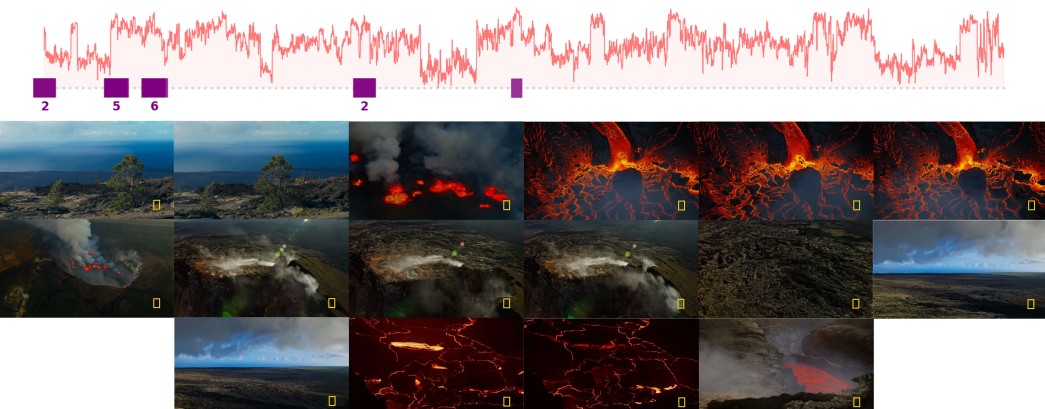

Figure 8: Frame selection comparison on a volcanic formation video. Uniform Sampling, CLIP (Top-K, K=16), and *A.I.R.* for the query about Nahuku's formation, with similarity score graphs showing *A.I.R.*'s focused sampling around relevant lava segments.

Question: Which of the following descriptions of the heroine's daily activities in a day is correct in chronological order?
A. Shopping, doing laundry, making the bed.,   B. Online shopping, making the bed, doing laundry.,   **C. Buy coffee, shop, make her bed.,**   D. None of the above.

## Uniform Sampling

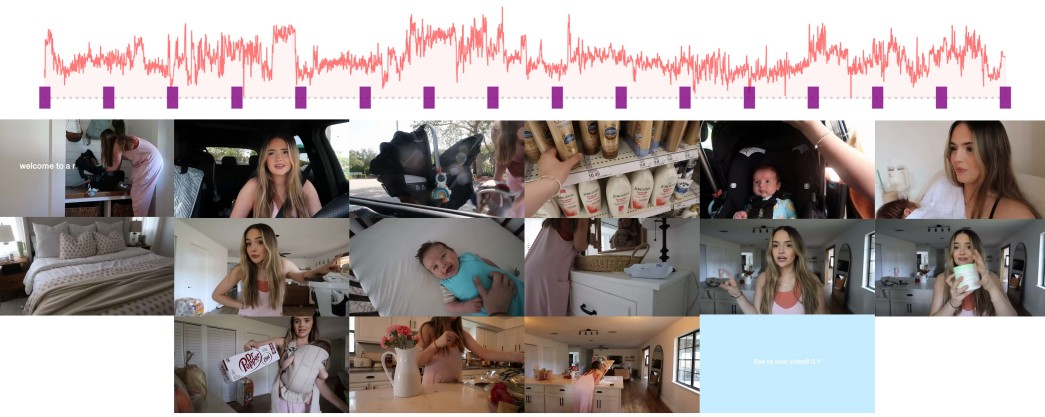

## CLIP (Top-K)

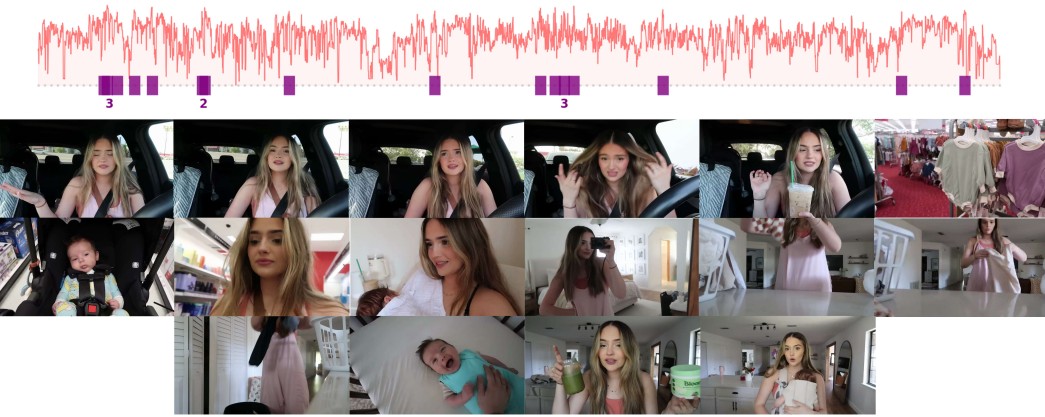

## A.I.R.

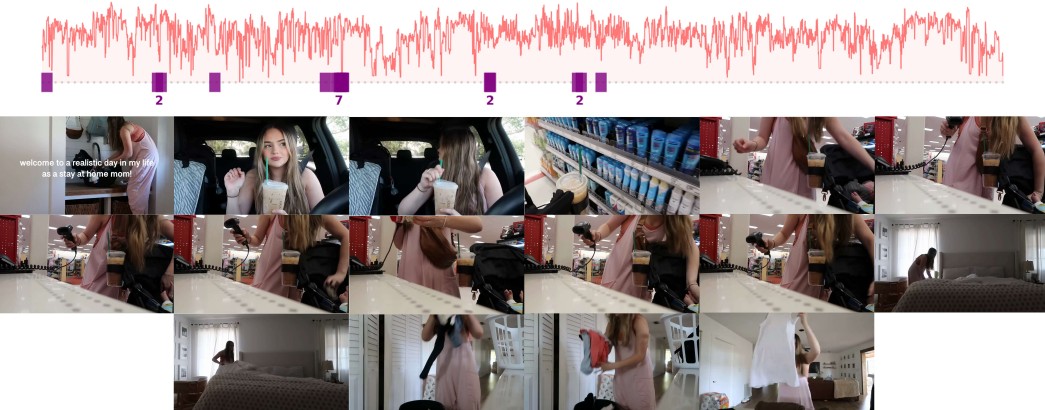

Figure 9: Additional frame selection on a daily routine video comparing Uniform Sampling, CLIP (Top-K, K=16), and *A.I.R.* for chronological activity ordering. *A.I.R.* effectively excludes numerous irrelevant frames while accurately capturing the complete sequence of key activities (buy coffee, shop, make her bed, do laundry).

