# OpenReview forum: "A.I.R.: Enabling Adaptive, Iterative, and Reasoning-based Frame Selection For Video Question Answering"
_ICLR.cc/2026/Conference — ICLR 2026 Poster_

### Official Review · Reviewer_iMeg · 2025-10-20

**Soundness:** 4
**Presentation:** 3
**Contribution:** 4
**Rating:** 6
**Confidence:** 5

**Summary:**

The paper proposes A.I.R., a training-free Adaptive, Iterative, and Reasoning-based framework for frame selection in VideoQA. It first adaptively samples query-relevant events based on similarity distributions, then iteratively refines frame selection using VLM reasoning within a cost-effective loop. Experiments show superior efficiency and accuracy across benchmarks.

**Strengths:**

The paper is original in introducing a reasoning-based, training-free iterative framework for frame selection, addressing the cost–accuracy trade-off in VideoQA. The methodology is solid and well-motivated, combining adaptive sampling with iterative refinement. Results show significant efficiency gains, and the paper is clear and impactful, improving VLM applicability to real-world video understanding.

**Weaknesses:**

The paper lacks runtime comparisons with existing frame selection methods, needs stronger justification of iterative VLM selection, and does not discuss generalization to other video-language tasks.

**Questions:**

1. The proposed rule-based frame selection achieves higher recognition accuracy, but the paper does not compare computational efficiency with existing frame selection methods (e.g., frame selection time).

2. The key contribution is the VLM-based iterative frame selection algorithm, yet the correctness of the iterative process requires further clarification. Additionally, it is unclear how performance compares if all frames are selected in a single, non-iterative pass using the same VLM under the maximum budget.

3. Essentially, the paper introduces another key-frame selection method for multimodal (video-language) understanding to improve accuracy. It remains unclear whether this approach generalizes to other video-language tasks, such as video-language retrieval.

---

> ### Author Response · Authors · 2025-11-23
> **Response to Reviewer iMeg - 1**
>
> Thank you for the positive evaluation and the insightful questions. We appreciate your recognition of our work's originality and impact, and are happy to address your specific concerns.
>
> ---
>
> ### Q1: Computational Efficiency Comparison (Question 1)
>
> We thank the reviewer for this important observation. We have provided comprehensive efficiency comparisons across **three key dimensions**: frame selection overhead, VLM inference time, and LLM call frequency.
>
> **1. Frame Selection Overhead (Algorithmic Efficiency):**
> Our manuscript (Table 7 in main paper) reports that A.I.R.'s total overhead for "Adaptive Initial Sampling" (0.03s) and "Iterative Frame Selection" (0.18s) is only **0.21 seconds**. Following your suggestion, we consulted related work and found:
>
> - **MDP3**: ~1.30s (ICCV25, from their Appendix B.6)
> - **Q-Frame**: ~0.30s (ICCV25, from their Appendix B.4)
> - **Frame-Voyager** : ~0.37s (ICLR25, from their Appendix C)
> - **BOLT**: ~1.24s (CVPR25, from their Appendix B)
> - **A.I.R. (Ours)**: **0.21s** （Best）
>
> This demonstrates that A.I.R.'s selection algorithm is **1.8-6.2× faster** than competing methods while achieving superior accuracy.
>
> **2. VLM Inference Time (End-to-End Efficiency):**
> To provide a direct computational comparison, we measured the end-to-end inference time of A.I.R. against agent-based methods using 7B-scale models on the Video-MME benchmark.
>
> **Table:** Computational Time Comparison on **Video-MME**. All compared methods use VLM under 8B.
>
> | **Method** | **Time (s)** |
> |------------|--------------|
> | Video-RAG | 353.6 |
> | Vgent | 66.8 |
> | MemVid | 57.3 |
> | **A.I.R.** ($w_{\text{worst}}=72$) | **42.3** |
> | **A.I.R.** ($w_{\text{worst}}=32$) | **21.9** |
>
> This demonstrates that A.I.R. achieves substantially lower computational costs than agent-based methods while maintaining superior performance, with our worst-case configurations providing **26.2–61.8%** speedup over the best competing approaches.
>
> **3. LLM Call Frequency (Agent-Based Methods Comparison):**
> Unlike agent-based methods requiring extensive multi-step reasoning, A.I.R. minimizes VLM calls through efficient iterative processing (Evaluated in **EgoMem** dataset):
>
> | **Method** | **Avg. LLM Calls** | **Efficiency Gain** |
> |------------|-------------------|---------------------|
> | VideoAgent (GPT-4) | 14.1 | **3.8× more than A.I.R.** |
> | VideoTree (GPT-4) | 9.6 | **2.6× more than A.I.R.** |
> | VideoLucy (DeepSeek-R1) | 6.3 | **1.7× more than A.I.R.** |
> | **A.I.R.** ($w_{\text{worst}}=72$) | **3.7** | **Most Higher Acc.** |
> | **A.I.R.** ($w_{\text{worst}}=32$) | **2.7** | **Most Efficient** |
>
> **4.Summary:**
> A.I.R. achieves **superior efficiency across all metrics**: fastest frame selection (0.21s), lowest VLM inference time (21.9-42.3s), and minimal LLM calls (2.7-3.7), while maintaining state-of-the-art accuracy. We will add Table 9 and explicit frame selection time comparisons to the revised manuscript.

---

> ### Author Response · Authors · 2025-11-23
> **Response to Reviewer iMeg - 2**
>
> ### Q2: Iterative Process Correctness (Question 2)
> Thank you for this excellent question related to core design of A.I.R. We provide both qualitative and quantitative evidence to demonstrate the necessity of our iterative process.
>
> **Qualitative Evidence: Figure 3 Case Study**
>
> Figure 3 provides a complete demonstration of A.I.R.'s two-stage workflow and how it correctly recovers ground-truth frames. **(1) Figure 3(a)** illustrates the **Adaptive Initial Sampling stage**, where our GMM-based adaptive thresholding identifies query-relevant temporal segments from CLIP similarity scores, and Event-Wise Sampling allocates frames proportionally across detected events to create a high-potential initial candidate set. **(2) Figure 3(b)** then demonstrates the **Iterative Frame Selection stage** using the Buddhist Temple query example. Initial CLIP-based sampling assigns a low score (0.4) to the correct temple frame while giving higher scores to contextually related Tofu-making frames. The iterative process correctly recovers the answer through systematic refinement. The VLM first analyzes high-CLIP-score candidates and validates a Tofu frame as contextually relevant, recognizing its relationship to the query's temporal structure ("After introducing Tofu making, what..."). This validation triggers **Localized Density Sampling** to explore the temporal neighborhood around the validated frame to discover the previously low-scored Buddhist Temple frame. Subsequent VLM analysis validates this frame as the definitive answer, recovering the ground truth that CLIP missed due to weak keyword associations.
>
> Additional workflow demonstrations in **Appendix Figure 7** show similar correct convergence patterns across diverse query types. Figures 3 & 7 demonstrate the framework's effectiveness: Adaptive Initial Sampling ensures broad coverage of high-potential regions, while Iterative Frame Selection refines them through semantic validation and temporal exploration
>
> **Quantitative Evidence: Ablation Study**
>
> In response to your specific question, we conducted the exact experiment you suggested: comparing A.I.R. against a **single-pass VLM baseline** that removes the iterative loop. This baseline uses only Adaptive Initial Sampling to obtain 32 candidate frames, then feeds all frames to the VLM in a single pass without iterative refinement.
>
> **Updated Ablation Table 4** (Row 6 highlighted):
>
> | # | Method | Avg. Frames | Acc. |
> |---|--------|-------------|------|
> | 1 | Uniform Sampling | 32.0 | 65.6 |
> | 2 | **A.I.R.** (Full method) | 24.8 | **68.2** |
> | 3 | w/ Fixed 32 frames (no Ada. Budget) | 32.0 | 68.3 |
> | 4 | w/o Adaptive Similarity Thresholding | 25.5 | 67.3 |
> | 5 | w/o Adaptive Initial Sampling | 25.1 | 66.9 |
> | **6** | **w/o Iterative Frame Selection (32 Frames)** | **32.0** | **65.2** |
> | 7 | w/o Interval Potential Ranking | 26.2 | 66.7 |
> | 8 | w/o Reasoning-based VLM Analysis | 32.0 | 66.0 |
> | 9 | w/o Localized Density Sampling | 24.5 | 67.2 |
>
>
> **Correctness Validation:**
>
> The ablation results validate three aspects of our iterative process's correctness. (1) **the iteration improves upon single-pass analysis** (Row 6 vs. Row 2: +3.0%), demonstrating that the iterative refinement systematically enhances frame quality rather than introducing noise or instability. (2) **VLM reasoning makes correct judgments** (Row 8: -2.2% without it), confirming that the VLM's semantic validation accurately identifies truly relevant frames beyond CLIP's keyword matching. (3) **LDS discovers correct adjacent frames** (Row 9: -1.0% without it), validating that temporal exploration around VLM-validated anchors successfully uncovers ground-truth frames through principled local search rather than random sampling. Together, these results prove that our iterative mechanism operates correctly as formalized in Algorithm 1.

---

> ### Author Response · Authors · 2025-11-23
> **Response to Reviewer iMeg - 3**
>
> ### Q3: Generalization to Other Video-Language Tasks (Question 3)
> This is an important question about the broader applicability of A.I.R., and we are pleased to report that our method successfully generalizes beyond video question answering to other video-language understanding tasks, which are already updated in the manuscript.
>
> **Extension to Temporal Grounding:**
>
> To demonstrate A.I.R.'s versatility, we evaluated our method on the **Charades-STA temporal grounding benchmark**, which is also known as the moment retrieval task (from TRACE [1]). Instead of answering questions, temporal grounding requires localizing specific moments in videos described by natural language queries.
>
> **Results on Charades-STA:**
>
> | **Model** | **R1@0.3** | **R1@0.5** | **R1@0.7** | **mIoU** |
> |-----------|------------|------------|------------|----------|
> | **Trained Temporal Grounding VideoLLMs** |||||
> | VTimeLLM | 51.0 | 27.5 | 11.4 | 31.2 |
> | HawkEye | 50.6 | 31.4 | 14.5 | 33.7 |
> | TimeChat | - | 32.2 | 13.4 | 30.6 |
> | TimeSuite | 69.9 | 48.7 | 24.0 | - |
> | TRACE | - | 40.3 | 19.4 | - |
> | **General VideoLLMs** |||||
> | GPT-4o | 55.0 | 32.0 | 11.5 | 35.4 |
> | Qwen2.5-VL-7B | 44.5 | 30.3 | 15.2 | 30.1 |
> | LongVA-7B-DPO | 22.6 | 10.1 | 2.2 | 14.6 |
> | Aria | 39.0 | 18.6 | 6.6 | 26.7 |
> | **Trained Frame Selection Methods** |||||
> | GenS (Aria) | 62.9 | 38.7 | 15.2 | 38.0 |
> | **Training-Free Frame Selection Methods** |||||
> | **A.I.R. (Qwen2.5-VL-7B)** | **59.5** | **39.5** | **18.0** | **38.8** |
>
> **Key Findings:**
>
> 1. **State-of-the-Art Among Training-Free Methods:** A.I.R. achieves the best results among all training-free approaches on R1@0.5 (39.5%), R1@0.7 (18.0%), and mIoU (38.8%), demonstrating that our adaptive frame selection mechanism naturally extends to temporal localization tasks.
>
> 2. **Competitive with Trained Methods:** A.I.R. outperforms the training-based GenS method at stricter IoU thresholds (R1@0.7: 18.0% vs. 15.2%; mIoU: 38.8% vs. 38.0%) while requiring **no task-specific training**. This is remarkable given that GenS is specifically trained for temporal grounding.
>
> 3. **Surpasses Specialized Temporal Grounding VideoLLMs:** A.I.R. outperforms dedicated temporal grounding models like HawkEye (R1@0.5: 39.5% vs. 31.4%) and TimeChat (R1@0.7: 18.0% vs. 13.4%), which are explicitly designed and trained for this task.
>
> 4. **Significant Improvement Over Base Model:** Compared to the base Qwen2.5-VL-7B (R1@0.5: 30.3%, mIoU: 30.1%), A.I.R. provides substantial gains (+9.2% on R1@0.5, +8.7% on mIoU), demonstrating that our frame selection strategy is highly beneficial for temporal grounding.
>
> **Future Directions:**
>
> While we have demonstrated generalization from video QA to temporal grounding, we plan to expand our method to additional tasks. Specifically, we will integrate A.I.R. into existing agent-based long-form video understanding systems, where it can provide enhanced extraction of video-relevant content for downstream tasks such as dense video captioning, action detection, and video-text retrieval.
>
>
> If there are any remaining questions or if additional experiments would strengthen our work further, we are fully committed to addressing them. Thank you again!
>
> Sincerely,
> The Authors
>
> [1] TRACE: Temporal Grounding Video LLM via Causal Event Modeling, ICLR 2025

---

> > ### Comment · Reviewer_iMeg · 2025-11-26
> >
> > Despite the authors' partial response to my concerns and my consideration of other reviews, I maintain my rating.

---

> ### Author Response · Authors · 2025-11-26
> **Response to Reviewer iMeg - Further Clarification**
>
> Thank you for your continued engagement. We have carefully reviewed your previous comments and believe our rebuttal addressed each concern raised. Regarding your note on a "partial response," could you kindly specify which particular aspects you feel remain unaddressed? We are eager to provide any further clarification needed to ensure your concerns are fully resolved.
>
> Regarding generalization, we prioritized Video Grounding (also recognized as a moment retrieval task) in Charades-STA in our initial response because it enables rigorous, direct comparison with established baselines like GenS [1]. In contrast, most prior frame selection works (e.g., BOLT, Q-Frame, MDP3, VideoLucy) focus exclusively on VideoQA, making Video Retrieval benchmarks scarce for this specific domain.
>
> [1] Generative Frame Sampler for Long Video Understanding, ACL Findings.

---

### Official Review · Reviewer_oTZs · 2025-10-26

**Soundness:** 2
**Presentation:** 3
**Contribution:** 2
**Rating:** 4
**Confidence:** 4

**Summary:**

This paper presents a training-free framework to improve accuracy in long-form VQA. The method starts with the adaptive initial sampling by using CLIP-based query-frame similarity and Gaussian mixture modeling, followed by an iterative reasoning loop that selectively applies a foundation VLM to high-potential frames. Experiments on several benchmarks show that the proposed method consistently improves accuracy.

**Strengths:**

The paper is well-written and easy to follow, with clear motivation, framework design, and experimental presentation.

The three-stage pipeline is also technically coherent and logically structured. The approach is training-free, which is good for integration into various VLMs,

The experiments and ablation studies demonstrate consistent improvements across multiple benchmarks.

**Weaknesses:**

While the paper is technically solid, its novelty and conceptual contribution are limited. The proposed method offers an effective improvement by integrating various schemes, but the overall contribution is incremental, because it only refines existing query-based frame-selection paradigms rather than offering a fundamental advance in long-video reasoning.

The paper does not include comparisons or discussions with agent-based methods, which represent another important line of work in long-video understanding.

The proposed method remains query-specific and cannot support multi-turn or persistent reasoning across video contexts.

**Questions:**

In addition to questions in weakness part, the authors are encouraged to address the following issues.

The iterative refinement uses several manually tuned hyperparameters (e.g., γ, α, β), and it's better to discuss their generalization across datasets.

It's suggested to discuss the future extensions, especially its combination with agent-based long-form video understanding systems.

---

> ### Author Response · Authors · 2025-11-23
> **Response to Reviewer oTZs**
>
> Thank you for your constructive feedback and thoughtful questions about our method's generalization and future directions.
>
> ---
>
> ### Q1: Incremental Contribution (Weakness 1)
>
> We respectfully clarify that A.I.R. shifts the frame selection paradigm from static, retrieval-based ranking to **dynamic, reasoning-guided search**. While integrating effective schemes, our core contribution is a systematic **Training-Free framework** that fundamentally resolves the efficiency bottleneck in long-video understanding. Unlike prior methods that require expensive specific training or exhaustive scanning, A.I.R. enables generic VLMs to comprehend long videos with **superior efficiency** (achieving ~4× speedup and minimal API costs). We further elaborate on this in our **General Response to Novelty Contribution: Heuristics vs. Principled Design**.
>
> ### Q2: Comparisons with Agent-based Methods (Weakness 2)
>
> We have thoroughly addressed this concern in our **General Response to Comparison with Agent-based State-of-the-Art Methods & Baselines**. If you have any other questions, we are more than happy to continue addressing them.
>
> ### Q3: Multi-Turn and Persistent Reasoning (Weakness 3)
> We acknowledge that A.I.R. is a query-specific framework. However, it is important to note that **most existing frame selection methods are inherently incompatible with persistent reasoning**, as they rely on query-relevance to filter vast video content. While this means inference time scales linearly with the number of queries, we argue that this design is the optimal choice for high-performance video understanding:
>
> 1. **Noise Reduction**: Methods that enable persistent reasoning often must retain massive context windows, which inevitably introduces significant irrelevant frames ("noise"). This information dilution degrades the VLM's reasoning capability. In contrast, by regenerating the frame set for each query, A.I.R. ensures high performance by strictly feeding the VLM only the most relevant evidence.
> 2. **Efficiency Makes it Practical**: Although our approach runs per-query, our superior efficiency mitigates the linear cost. A.I.R. is fast enough to be run dynamically for each turn in a conversation without prohibitive latency, offering a practical balance between high-performance and efficiency.
>
> ### Q4: Hyperparameter Generalization Across Datasets (Question 1)
>
> We have comprehensively addressed this concern in our **General Response to Robustness and Generalization of Hyperparameters**. If you have any other questions, we are more than happy to continue answering them.
>
> ### Q5: Future Extensions to agent-based long-form video understanding system (Question 2)
>
> This is an insightful suggestion for future work. We first clarify that we have already conducted comprehensive comparisons against recent agent-based methods in our revised manuscript (**Table 6 in Appendix A4.1**), where A.I.R. achieves superior performance. Looking forward, A.I.R. naturally complements agent-based systems through its **plug-and-play design**. Our Adaptive Initial Sampling and Iterative Frame Selection stages can serve as an intelligent front-end module that dynamically provides high-quality frame sets to downstream agent reasoning components, replacing their current fixed-sampling or exhaustive-search strategies. This integration would be particularly valuable when agents perform diverse long-form video tasks beyond question answering, such as temporal grounding, object detection, or multi-step reasoning, where our method could work synergistically with task-specific modules (e.g., grounders, detectors) as an efficient video information provider. We have already validated A.I.R.'s generalization capability to temporal grounding tasks in our manuscript (starting from L505 in the **Generalization to Grounding Task Analysis section**), where our training-free approach achieves competitive performance with specialized grounding models on Charades-STA, demonstrating the framework's extensibility to broader video understanding scenarios that agent-based systems typically address.
>
>
> If there are any remaining questions or if additional experiments would strengthen our work further, we are fully committed to addressing them. Thank you again!
>
> Sincerely,
> The Authors

---

> > ### Comment · Reviewer_oTZs · 2025-11-26
> >
> > I appreciate the authors’ detailed rebuttal. The comparison with agent-based methods is informative, and it is impressive that the proposed method is comparable to these agent-based approaches. The clarifications regarding multi-turn and persistent reasoning and hyperparameter generalization are also reasonable.
> >
> > However, my concern regarding novelty still partially remains. While the framework is well designed and demonstrates practical value, the contribution is still within the query-based frame selection paradigm. This direction has inherent limitations and has been extensively explored in prior work. So, the method looks more incremental than fundamentally innovative.
> >
> > This paper is technically sound and well-designed, but from a conceptual perspective, the paper may not advance beyond the frame selection paradigm the inherently has some limitations. Given this, I will maintain my original rating for now. If other reviewers or the AC find that the paper offers sufficient novelty, I would not oppose acceptance.

---

> > > ### Author Response · Authors · 2025-11-26
> > > **Response to Reviewer oTZs - Further Clarification**
> > >
> > > We sincerely thank the reviewer for the thoughtful discussion and for recognizing the technical soundness, practical value, and competitive performance of our approach—especially in comparison to recent agent-based methods.
> > >
> > > Regarding the concern about novelty within the frame-selection line of work, we agree that the high-level paradigm is established. However, recent progress (e.g., BOLT (CVPR’25), AKS (CVPR’25), VideoLucy (NeurIPS’25), MDP3 (ICCV’25)) underscores that query-conditioned frame selection remains a central and actively developing direction for long-video understanding: it explicitly targets long-video redundancy and the context-length/cost bottleneck of MLLMs by extracting the most informative evidence for a given query. Thus, while the paradigm has known limitations (e.g., query-dependence), it is still significant for making long-video MLLM reasoning practical and scalable.
> > >
> > > Our contribution is to advance this paradigm by addressing two underexplored but practically limiting issues in existing pipelines: (1) the low affinity of CLIP-style retrieval to complex, compositional queries, and (2) the computation explosion that arises when applying VLM-based reasoning naïvely over many frames. A.I.R. introduces a training-free, adaptive, iterative selection framework that (i) supports balanced query-specific sampling via adaptive initial sampling, (ii) leverages VLM reasoning to validate and refine candidate evidence rather than relying solely on embedding similarity, and (iii) maintains the superior efficiency via the iterative algorithm while enabling re-discovering missing information in unsampled frames. Importantly, our results show that this design can close much of the capability gap to agentic systems while preserving the efficiency advantages of selection-based approaches—making it a pragmatic step forward rather than a minor modification.
> > >
> > > Finally, to demonstrate that the framework is generalizable, we additionally evaluate on video grounding (see **Response to Reviewer iMeg - 3**) and achieved SOTA performance compared with other training-free methods, which proves the generalization of our proposed approach beyond VideoQA.
> > >
> > > We appreciate your openness to acceptance if the AC and other reviewers deem the novelty sufficient. If there are remaining concerns, we would be happy to further clarify and address them, and we hope you will reconsider your rating in light of these points.

---

### Official Review · Reviewer_jkbb · 2025-11-01

**Soundness:** 3
**Presentation:** 4
**Contribution:** 3
**Rating:** 6
**Confidence:** 4

**Summary:**

The paper proposes A.I.R., which is a multi-stage frame selection heuristics for long video question answering. The pipeline emphasizes efficiency by minimizing the number of frames fed to the heavy-weight reasoning-based VLM for analysis. The pipeline first computes question-frame similarity scores using a light-weight CLIP-style VLM, and then perform multi-stage selection to determine the frames for the expensive MLLM-powered question answering. The selection process include an initial adaptive sampling step, where scores are modeled as a 2-component GMM (relevant and irrelevant), followed by a loop of interval potential ranking, reasoning-based VLM analysis, early stop mechanism and localized density sampling to add or remove from the candidate frame set until a fixed budget is used. The frames are then input to a MLLM with the question to generate the answer. The method is evaluated on a wide range of models (VILA-1.5, QwenVL-2.5, InternVL-3, LLaVA-OneVision) and on multiple datasets (VideoMME, MLVU, LVB, EgoSchema, NextQA), showcasing its effectiveness in a broad range of settings.

**Strengths:**

* The paper addresses the topic of long-video understanding, which is of high real-world value and is essential for a broad set of video analysis applications.

* By demonstrating the effectiveness of human-derived heuristics, the paper reveals key limitations of recent LLM-agent-based frame selection pipelines, which often incur much higher costs due to frequent calls to expensive LLMs/MLLMs.

* The paper is very well written, with a clear framework and a good coverage of technical details.

* The proposed method is evaluated against a wide range of recent methods across multiple datasets and shows strong performance, demonstrating its efficiency and effectiveness in a comprehensive way. Ablation experiments also demonstrate the effectiveness of individual components of the proposed method.

**Weaknesses:**

* The CLIP affinity score is used extensively in multiple stages of the pipeline, which raises questions about how the quality of this score may affect the overall efficiency and accuracy of the pipeline. CLIP models are known to have several limitations, such as a relatively low token number limits of the text encoder (tens to hundreds, versus thousands or more in MLLMs), being less capable at modeling complex relationships in text expressions [1], and being less capable at extracting fine-grained visual information [2]. A low precision or recall may result in excessive frames being sent to reasoning VLM or information loss, potentially impacting the efficiency or quality of the pipeline. A more detailed discussion [e.g., see questions (a, b)] on the issue might clarify the strength and limitations of the proposed approach.

* In Table 2 and 3, it seems that the performance gain of frame selection is closing rapidly as the backbone model quality improves, which raises some concerns about the long-term impact of the work or the frame selection pipelines in general.


[1] Yuksekgonul, Mert, et al. "When and Why Vision-Language Models Behave like Bags-Of-Words, and What to Do About It?." The Eleventh International Conference on Learning Representations.

[2] Xie, Chunyu, et al. "FG-CLIP: Fine-Grained Visual and Textual Alignment." Forty-second International Conference on Machine Learning.

**Questions:**

* Following weakness 1, have the authors encountered any failure cases of the CLIP text encoder, e.g., questions longer than the max number of tokens or CLIP failing to capture the semantic of some complex expressions?

* Following question 1, if systematic failure cases exist, could the paper include a discussion about how they are mitigated by designs in the proposed pipeline, or how they could possibly be mitigated in future works?

* Following weakness 2, if resources permit, could the paper include more results on larger or closed models to better demonstrate its advantages of being training-free and avoiding large numbers of MLLM calls?

---

> ### Author Response · Authors · 2025-11-22
> **Response to Reviewer jkbb - 1**
>
> Thank you for the thorough review and the constructive questions. We appreciate your recognition of our work's strengths and are happy to address your specific concerns.
>
> ---
>
> ### Q1: CLIP Failure Cases Analysis (Weakness 1 & Question 1)
>
> We sincerely thank the reviewer for highlighting the limitations inherent in CLIP-based models. We agree that this is a critical motivation for our work and have updated the introduction to include the suggested citations. In our experiments, we identified systematic failure cases where CLIP’s reliance on surface-level features negatively impacts initial frame selection. We explicitly document these cases to demonstrate the necessity of our iterative VLM-based refinement.
>
> **Detailed Analysis of Failure Cases:**
> In the main paper, we present the **Buddhist Temple Example** (Figure 1(b) & Figure 3(b)) using the query: *"After introducing Tofu making, what kind of traditional technique or scenic spot did the youtuber introduce...?"*
> * **Semantic Ambiguity (Figure 1(b)):** CLIP fails to parse the complex temporal condition ("After introducing..."). Instead, treating the query as a "bag of keywords," it assigns high similarity scores (0.7-0.8) to frames containing "Tofu" or visually similar contexts like a "Market." Conversely, the ground-truth frame showing a "Buddhist Temple" receives a low score (0.4) due to a lack of direct keyword overlap, despite being the semantically correct answer.
> * **Impact on Sampling (Figure 3(a)):** This inaccurate scoring directly propagates to the Adaptive Initial Sampling stage. Because the *Buddhist Temple* frames are assigned low scores, they are excluded from the initial candidate pool. This necessitates the proposed Iterative Frame Selection (discussed in **Q2**), which effectively recovers these missing keyframes through reasoning-based expansion.
> * **Additional Evidence (Figure 7):** Furthermore, in **Figure 7** (Appendix, page 26), we analyze a "Life Record" video with the query: *"What did the male protagonist... do immediately after finishing his personal report?"* Here, CLIP-based sampling captures only a generic frame of *cooking in the kitchen*, failing to retrieve the specific context required to infer the action *having lunch*.
>
> **Summary of Limitations:**
> These cases underscore two fundamental deficits in CLIP-based retrieval:
> * **Limited Understanding of Complex Queries:** CLIP struggles with compositional semantics, often prioritizing keyword matching over holistic understanding.
> * **Inability to Handle Temporal Relations:** It lacks the capacity to reason about temporal logic (e.g., "immediately after"), causing it to undervalue frames that are temporally relevant but visually distinct from the query keywords.
>
> Regarding the **text token length limit of CLIP**, we conduct a detailed ablation study on different CLIP backbones in **Table 5**. A comprehensive analysis of these results is also provided in **Q2**.

---

> ### Author Response · Authors · 2025-11-22
> **Response to Reviewer jkbb - 2**
>
> ### Q2: Mitigation Strategies in A.I.R. (Weakness 1 & Question 2)
>
> A.I.R. mitigates CLIP's limitations through **two synergistic designed steps**: Reasoning-based VLM Analysis and Localized Density Sampling. We provide both qualitative and quantitative evidence below.
>
> **Mitigation Method: Synergistic VLM Reasoning and Localized Density Sampling**
> The core insight is that CLIP provides a coarse-grained initial filter based on visual-textual similarity, but lacks the semantic reasoning capability to understand complex query-frame relationships. A.I.R. addresses this limitation through **two synergistic steps**. **First, VLM reasoning** performs step-by-step analysis that validates semantic correctness beyond keyword matching—when CLIP assigns low scores to frames due to weak surface-level associations, the VLM can recognize contextual relevance and temporal dependencies that CLIP misses. **Second, Localized Density Sampling (LDS)** (Eq. 5) exploits temporal coherence by performing dense sampling around VLM-validated frames, uncovering temporally adjacent frames that were initially assigned low CLIP scores but contain semantically relevant information. Together, these components create a **feedback loop** (Alg. 1) that forms a powerful recovery mechanism, which ensures that even when CLIP fails, the system can ultimately uncover the correct frames that CLIP initially excluded.
>
> **Qualitative Evidence: Pipeline Process Analysis（Figure 3(b) & 7）**
> Following the Figure 3(a)'s failure of CLIP, Figure 3(b) demonstrates this iterative frame selection's "rediscover" ability in the Buddhist Temple example. CLIP assigns a low score (0.4) to the correct answer frame, which would lead to its exclusion in CLIP-only methods. However, A.I.R.'s VLM Analysis validates a contextually related frame showing "Tofu making" (scored 4/5 by the VLM), recognizing it as part of the video's narrative sequence. This validation triggers Localized Density Sampling (LDS) to explore the temporal neighborhood, where the VLM subsequently discovers and validates the Temple frame through semantic understanding of the query's temporal structure ("After introducing Tofu making, what..."). The detailed workflow analyses in Appendix Figure 7, where our VLM analysis discover the temporal relationship between *cooking meal* and *have lunch*, and thus sampling more related frames around. This example further demonstrates our method's effectiveness across diverse query types, showing how VLM reasoning consistently recovers frames that CLIP undervalues.
>
> **Quantitative Evidence: Component Ablation and CLIP Variant Analysis**
>
> We conducted two complementary experiments to validate our mitigation method. First, we ablated key components to isolate their contributions (Table 4). Second, we tested five CLIP variants to verify robustness across different text encoders (Table 5).
>
> **Table 4:** Component Ablation on Video-MME (w/o subtitle, InternVL3-8B)
>
> | # | Method | Avg. Frames | Acc. |
> |---|--------|-------------|------|
> | 1 | Uniform Sampling (baseline) | 32.0 | 65.6 |
> | 2 | **A.I.R. (full method)** | **24.8** | **68.2** |
> | 6 | w/o Iterative Frame Selection (CLIP only) | 32.0 | **65.2** |
> | 8 | w/o Reasoning-based VLM Analysis | 32.0 | 66.0 |
> | 9 | w/o Localized Density Sampling | 24.5 | 67.2 |
>
> **Table 5:** CLIP Model Ablation on Video-MME (w/o subtitle, InternVL3-8B, max 32 frames)
>
> | # | CLIP Model | Max Tokens | Acc. |
> |---|------------|------------|------|
> | 1 | Uniform Sampling (no CLIP) | - | 65.6 |
> | 2 | A.I.R. + CLIP-ViT-B | 77 | 66.8 |
> | 3 | **A.I.R. + EVA-CLIP-L (Ours)** | **77** | **68.2** |
> | 4 | A.I.R. + LongCLIP-L | **248** | 67.4 |
> | 5 | A.I.R. + CLIP-ViT-L | 77 | 67.8 |
> | 6 | A.I.R. + SigLIP-large | 64 | 67.1 |
>
> **Key Observations:**
> * **(1) CLIP-Only Sampling Fails:** Without Iterative Frame Selection (Row 6, Table 4), performance drops to 65.2%, **below the uniform sampling baseline** (65.6%), confirming that CLIP alone is insufficient when queries require semantic reasoning beyond surface-level similarity.
> * **(2) VLM Reasoning is Critical:** Removing Reasoning-based VLM Analysis (Row 8) causes a -2.2% drop to 66.0%, demonstrating that deep semantic validation is the **primary driver** in overcoming CLIP's limitations and recovering undervalued frames.
> * **(3) LDS Enables Temporal Recovery:** Removing LDS (Row 9) results in -1.0% drop to 67.2%, confirming that temporal exploration around VLM-validated frames uncovers critical adjacent content. The synergy between VLM reasoning and LDS accounts for the full +2.6% improvement.
> * **(4) Robustness Across CLIP Variants:** All CLIP variants (Table 5) achieve consistent improvements (+1.2% to +2.6%), with EVA-CLIP-L best at 68.2%. Notably, LongCLIP-L (248 tokens) underperforms EVA-CLIP-L (77 tokens) by -0.8%, showing that **visual-semantic alignment** matters more than text encoder capacity.

---

> ### Author Response · Authors · 2025-11-22
> **Response to Reviewer jkbb - 3**
>
> ### Q3: Results on Larger and Closed Models (Weakness 2 & Question 3)
>
> We appreciate this suggestion and have conducted comprehensive experiments across model scales, from 7B open-source models to 72B large models and frontier closed models (GPT-4o), to validate A.I.R.'s effectiveness and training-free advantages (Table 3 in manuscript).
>
> **Table 3:** A.I.R. Performance Across Model Scales on Video-MME (w/o subtitle, ≤32 frames)
>
> | **Model** | **Size** | **Baseline Acc.** | **+A.I.R. Acc.** | **Improvement** |
> |-----------|----------|-------------------|------------------|-----------------|
> | QwenVL-2.5 | 7B | 60.8 | **65.0** | **+4.2%** |
> | QwenVL-2.5 | 32B | 63.7 | **66.2** | **+2.5%** |
> | QwenVL-2.5 | 72B | 67.0 | **68.2** | **+1.2%** |
> | GPT-4o (Closed) | - | 61.8 | **65.1** | **+3.3%** |
>
> **Key Findings:**
>
> Our experiments across model scales (Table 3, also updated in manuscript) demonstrate two critical advantages of A.I.R.'s training-free framework. First, **model-agnostic effectiveness**: A.I.R. provides consistent improvements across diverse VLM scales—from 7B open-source models (+4.2%) to 72B large models (+1.2%) and closed APIs like GPT-4o (+3.3%)—confirming that our adaptive frame selection strategy generalizes across architectures and parameter counts without requiring model-specific tuning. Second, **efficiency guarantee under worst-case analysis**: Even when using large frontier models like GPT-4o, our iterative algorithm ensures that the worst-case VLM workload (Eq. 7) remains bounded below fixed-budget baselines (e.g., $w_{\text{worst}} = 72$ analyzed frames vs. 128 fixed frames in conventional methods), as formalized in Eq. 6. This theoretical guarantee, combined with our Early Stop Mechanism, means that A.I.R. maintains superior computational efficiency regardless of the underlying VLM's scale or cost, making it uniquely practical for leveraging expensive frontier models where minimizing API calls is critical.
>
> **Future Evaluations on Emerging Models:**
>
> We are committed to continuously validating A.I.R. on the latest models like **Claude 3.5 Sonnet** and **Gemini 2.5 Pro**, and other frontier multimodal models. These evaluations will be included in future revisions to further demonstrate A.I.R.'s sustained effectiveness as VLMs continue to improve.
>
>
> If there are any remaining questions or if additional experiments would strengthen our work further, we are fully committed to addressing them. Thank you again!
>
> Sincerely,
> The Authors

---

### Official Review · Reviewer_73LF · 2025-11-01

**Soundness:** 3
**Presentation:** 3
**Contribution:** 2
**Rating:** 4
**Confidence:** 4

**Summary:**

The paper tackles a core challenge in Video Question Answering (VideoQA) — how to efficiently select a small set of informative video frames that best correspond to a textual query. To overcome this trade-off, the authors propose A.I.R., a training-free framework for Adaptive, Iterative, and Reasoning-based frame selection. A.I.R. adaptively chooses the most relevant frames by combining efficient CLIP-based similarity analysis with targeted reasoning from a VLM. A.I.R. consists of two main stages, including adaptive initial sampling and iterative frame selection.  The method was extensively tested with various foundation VLMs, including InternVL-3, QwenVL-2.5, LLaVA-OneVision, and VILA-1.5, on both long-video and short-video benchmarks.

**Strengths:**

1. The authors provide a clear problem motivation, starting from the inefficiency of uniform sampling and the limitations of existing CLIP-based and VLM-based frame selection methods. The technical pipeline is presented logically, with consistent terminology and step-by-step exposition of each stage (Adaptive Initial Sampling → Iterative Frame Selection → QA).
2. This paper shows high-quality figures and conceptual illustrations. Figures such as Figure 1 (problem motivation), Figure 2 (overall pipeline), and Figure 3 (detailed visualization of both Adaptive Sampling and Iterative Selection stages) are visually rich and well-annotated, effectively guiding the reader through the method.
3. The technical part is given with many details. Readers can easily grasp the general idea of this paper.

**Weaknesses:**

1. Although the proposed A.I.R. framework achieves practical performance improvements, its core design is composed mainly of heuristic strategies rather than a principled methodological innovation. Many components—such as the adaptive thresholding via Gaussian Mixture Models, interval potential ranking, and localized density sampling—are intuitive extensions or empirical engineering choices rather than conceptually new formulations. As a result, the framework feels more like an optimized combination of existing ideas than a grounded advancement. This reliance on heuristics makes it difficult for readers to extract generalizable insights or new theoretical understanding from the paper, which somewhat diminishes its methodological contribution.
2. The proposed A.I.R. framework involves a large number of hyperparameters, which makes the overall method difficult to interpret and reproduce. This abundance of loosely motivated hyperparameters weakens the methodological transparency of A.I.R. and increases the barrier for readers to understand or extend the framework fully.
3. The performance advantage of the proposed A.I.R. framework is not particularly significant, and the experimental section lacks comprehensive comparisons with the most recent state-of-the-art methods. Although the paper reports moderate gains (often around 1–2% improvement on several benchmarks such as Video-MME and MLVU), these margins are relatively small given the additional complexity introduced by the multi-stage adaptive and iterative procedures. Moreover, the experiments primarily compare A.I.R. against earlier CLIP-based or training-free frame selection baselines (e.g., MDP3, BOLT, Frame-Voyager), but do not include newer or stronger contemporary methods that could provide a more convincing assessment of its competitiveness. Consequently, the empirical results do not clearly demonstrate that A.I.R. represents a substantial advancement over current techniques.

**Questions:**

The paper does not provide sufficient details or rationale regarding the hyperparameter settings used in the A.I.R. framework. How could this issue be addressed to improve the transparency and reproducibility of the proposed method?

---

> ### Author Response · Authors · 2025-11-22
> **Response to Reviewer 73LF**
>
> Thank you for your detailed feedback. We appreciate your constructive comments regarding novelty, reproducibility, performance and comparisons. We address your specific concerns below.
>
> ---
>
> ### Q1: Novelty Contributions (Weakness 1)
>
> We respectfully clarify that A.I.R. is not merely a combination of engineering heuristics, but a principled framework expressly designed to resolve the perception-computation trade-off in long-form video understanding. While individual components (such as GMM-based adaptive thresholding) are established, our core methodological innovation lies in reformulating frame selection as a dynamic, coarse-to-fine search problem rather than a static ranking task. These components do not function in isolation; they are integrated into a synergistic loop where reasoning actively guides sampling. We provide a comprehensive explanation of this design in **General Response to Novelty Contribution: Heuristics vs. Principled Design**.
>
>
> ---
>
> ### Q2: Hyperparameter Reproducibility (Weakness 2, Also as Question 1)
>
> We have comprehensively addressed this concern in our **General Response to Robustness and Generalization of Hyperparameters.** If you have any other questions, we are more than happy to continue addressing them.
>
>
> ---
>
> ### Q3: Performance Gains (Weakness 3.1)
>
> We respectfully disagree with the characterization of our performance gains as "modest." As shown in our results in Tables 1&2, our method achieves state-of-the-art performance across most benchmarks, delivering substantial improvements over strong baselines. Specifically, we boost QwenVL-2.5 by 8.2% on MLVU and 7.0% on NextQA, and enhance InternVL-3 by 6.1% on MLVU. Even on highly competitive benchmarks like Video-MME, we secure consistent gains of 4.2% (QwenVL-2.5) and 2.6% (InternVL-3). In the context of challenging competition in modern video understanding, where improvements are typically incremental, the performance gains of our method represent a significant advancement.
>
> Furthermore, these gains must be evaluated alongside our computational efficiency. The complexity of our framework is directly justified by its superior resource utilization. As detailed in Table 7, our method reduces the average frame analysis count from 128 to 36.5 compared to conventional VLM analysis methods, resulting in a 3.8$\times$ speedup (162.03s vs. 42.31s) without compromising accuracy. This advantage is even more pronounced in direct comparisons; for instance, while VideoTree requires processing 128 frames to achieve 75.6% on NextQA, our method utilizes only 32.2 frames to reach 82.6% (Table 6). This demonstrates that our approach does not merely add complexity; rather, it delivers a 7.0% accuracy improvement using ~4$\times$ fewer frames, offering a highly favorable trade-off between performance and efficiency.
>
>
> ---
>
> ### Q4: Comparison  newer or stronger Baselines (Weakness 3.2)
>
> We have thoroughly addressed this concern in our **General Response to Comparison with Agent-based State-of-the-Art Methods & Baselines.** If you have any other questions, we are more than happy to continue addressing them.
>
>
> If there are any remaining questions or if additional experiments would strengthen our work further, we are fully committed to addressing them. Thank you again!
>
> Sincerely,
> The Authors

---

### Author Response · Authors · 2025-11-22
**General Response**

Dear Reviewers, Area Chairs, and Program Chair,

We sincerely thank you all for considering our submission and for the insightful and constructive feedback provided. We greatly appreciate the time and effort each reviewer has dedicated to evaluating our work.

**Summary of Reviewers' Comments:** We are grateful for the reviewers' positive feedback. We were happy that the reviewers noted A.I.R.’s contributions:

* **Motivation and Significance.** (Reviewers **73LF**, **jkbb**, and **oTZs**).
    Reviewer **73LF** highlighted that the paper *"tackles a core challenge in Video Question Answering... how to efficiently select a small set of informative video frames,"* and praised the clear problem motivation. Reviewer **jkbb** acknowledged that the work addresses *"long-video understanding, which is of high real-world value."* Reviewer **oTZs** also noted the clear motivation in addressing the accuracy trade-offs.

* **Clarity and Presentation.** (Reviewers **73LF**, **jkbb**, **oTZs**, and **iMeg**).
    Reviewer **jkbb** stated the paper is *"very well written, with a clear framework,"* giving it an "excellent" rating for presentation. Reviewer **73LF** praised the *"high-quality figures and conceptual illustrations,"* specifically noting Figures 1, 2, and 3 as visually rich. Reviewer **oTZs** found the paper *"easy to follow,"* and Reviewer **iMeg** agreed the methodology is solid and well-motivated.

* **Efficiency and Performance.** (Reviewers **jkbb**, **oTZs**, and **iMeg**).
    Reviewer **iMeg** highlighted that *"Results show significant efficiency gains,"* and Reviewer **jkbb** noted that the method showcases *"strong performance, demonstrating its efficiency and effectiveness in a comprehensive way."* Reviewer **oTZs** confirmed that experiments demonstrate *"consistent improvements across multiple benchmarks."*

* **Comprehensive Evaluation.** (Reviewers **73LF** and **jkbb**).
    Reviewer **73LF** appreciated that the method was *"extensively tested with various foundation VLMs... on both long-video and short-video benchmarks."* Reviewer **jkbb** emphasized that the evaluation covers *"a wide range of models... and multiple datasets... showcasing its effectiveness in a broad range of settings,"* and further noted that ablation studies successfully demonstrate the effectiveness of individual components.

* **Methodological Design (Training-Free).** (Reviewers **oTZs**, **iMeg**).
    Reviewer **oTZs** appreciated that *"The approach is training-free, which is good for integration into various VLMs."* Reviewer **iMeg** recognized the originality in introducing a *"reasoning-based, training-free iterative framework."*

## Summary & Response of Common Concerns
In response to your valuable comments and suggestions regarding hyperparameters, novelty, and comparisons, we have conducted several new experiments, performed additional analyses, and introduced new baselines to enhance the robustness and comprehensiveness of our submission. These enhancements have been incorporated into both the main manuscript and the appendix (marked as **yellow texts**). Specifically, we summarize and respond to three common concerns below: **(1) Novelty Contribution:** A.I.R.'s principled design beyond heuristics; **(2) Robustness to Hyperparameters:** systematic ablations across 18 configurations; and **(3) SOTA Comparisons with Agent-based Methods:** benchmarking against 9 recent agent-based methods. Detailed responses follow below.

---

> ### Author Response · Authors · 2025-11-22
> **General Response to Novelty Contribution: Heuristics vs. Principled Design (Reviewers 73LF, oTZs)**
>
> **Concern:** Reviewer 73LF noted that A.I.R.'s core design consists mainly of heuristic strategies rather than principled methodological innovation, which "makes it difficult for readers to extract generalizable insights or new theoretical understanding." Reviewer oTZs similarly observed that the method "refines existing query-based frame-selection paradigms rather than offering a fundamental advance."
>
> **Response:** We respectfully disagree with the characterization that A.I.R. is merely a collection of heuristics. To address the problem of existing frame selection methods, which is the poor performance of lightweight models on complex queries and high computational cost of VLM-based analysis, we design a novel framework that focus on synergistic integration and adaptive application. Our key innovations include:
>
> **Novel Contributions:**
>
> 1. **Adaptive Initial Sampling Stage:** Rather than using fixed thresholds, we propose a GMM-based adaptive threshold that separates high-relevance and low-relevance frames for each video's unique distribution. We refine the events through Merging (combining nearby events) and Pruning (removing short noisy segments). Finally, **Event-Wise Sampling** ensures every event is represented while allocating more frames to longer, sustained events, yielding an initial sampling set focused on high-potential temporal regions.
>
> 2. **Iterative Frame Selection Stage:** To progressively refine the initial frames in a cost-effective manner, we propose a 4-step iterative algorithm. **Interval Potential Ranking** (Step 1) prioritizes high-potential temporal regions by scoring intervals rather than individual frames, providing a more comprehensive measure of temporal importance. **Reasoning-based VLM Analysis** (Step 2) validates candidates through generated textual justifications, ensuring only semantically relevant frames are retained. An **Early Stop Mechanism** (Step 3) terminates the process once the adaptive budget is met, preventing unnecessary computation. **Localized Density Sampling** (Step 4) dynamically discovers fine-grained frames around validated keyframes through exponentially growing sampling strides, creating a feedback loop that uncovers critical moments initially missed by CLIP-based sampling. This synergistic loop progressively converges on optimal frame selection through adaptive iteration.
>
> 3. **Systematic Framework and Efficient Performance:** We propose a systematic framework to address the limitations of existing methods, which are: lightweight models like CLIP fail to capture complex query semantics, while VLM-based analysis achieves higher accuracy but incurs prohibitive costs. A.I.R. addresses both limitations through a two-stage framework where CLIP first identifies high-potential frames, and VLM analysis works on small candidate batches per iteration. Our formal analysis proves A.I.R. achieves superior efficiency and high performance by strategically bounding VLM calls.

---

> ### Author Response · Authors · 2025-11-22
> **General Response to Robustness and Generalization of Hyperparameters (Reviewers 73LF, oTZs)**
>
> ### 2. Robustness and Generalization of Hyperparameters (Reviewers 73LF, oTZs)
> **Concern:** Reviewers 73LF and oTZs raised concerns about the large number of hyperparameters in A.I.R., questioning whether the framework is difficult to interpret, reproduce, and generalize across different video types and datasets.
>
> **Response:** We conducted a comprehensive systematic ablation study to demonstrate the robustness of A.I.R. across diverse hyperparameter configurations. We tested 18 different configurations by independently varying each key parameter: GMM coefficient ($\gamma$), minimum frame distance ($\alpha$), LDS growth factor ($\beta$), pruning threshold ($l_{\min}$), and merging gap ($d_{\min}$). Additionally, we evaluated four strategy-based combinations representing different video scenarios (Short-Video, Long-Video, Conservative, and Aggressive sampling strategies).
>
> **Key Finding: Strong Robustness and Consistent Performance.** All 18 configurations maintain performance within ±1.1% of our default setting (highlighted in gray), demonstrating that A.I.R. is highly robust to hyperparameter variations. On LongVideoBench, accuracy ranges from 61.7% to 62.9% (default: 62.8%), while on NextQA, accuracy ranges from 81.8% to 82.8% (default: 82.6%).
>
> **Hyperparameter Ablation Tables**: Systematic ablation study of **A.I.R.** hyperparameters using InternVL3-8B. Each table varies one parameter while keeping others at default values ($\gamma=0.7$, $\alpha=15$, $\beta=1.5$, $l_{\min}=20$, $d_{\min}=2$).
>
> ---
>
>
>
> **Table 1:** GMM Coefficient ($\gamma$)
>
> | # | $\gamma$ | LVB | NextQA |
> |---|----------|-----|--------|
> | **Default** | **0.7** | **62.8** | **82.6** |
> | 1 | 0.3 | 62.2 | 82.3 |
> | 2 | 0.5 | 62.6 | 82.6 |
> | 3 | 0.9 | 62.0 | 82.2 |
>
> **Table 2:** Min Frame Distance ($\alpha$)
>
> | # | $\alpha$ | LVB | NextQA |
> |---|----------|-----|--------|
> | **Default** | **15** | **62.8** | **82.6** |
> | 4 | 10 | 62.4 | 82.8 |
> | 5 | 20 | 62.8 | 82.1 |
> | 6 | 25 | 62.9 | 82.0 |
>
> **Table 3:** LDS Growth Factor ($\beta$)
>
> | # | $\beta$ | LVB | NextQA |
> |---|---------|-----|--------|
> | **Default** | **1.5** | **62.8** | **82.6** |
> | 7 | 1.2 | 62.4 | 82.4 |
> | 8 | 1.7 | 62.6 | 82.5 |
> | 9 | 2.0 | 62.1 | 82.1 |
>
> **Table 4:** Pruning Threshold ($l_{\min}$)
>
> | # | $l_{\min}$ | LVB | NextQA |
> |---|------------|-----|--------|
> | **Default** | **20** | **62.8** | **82.6** |
> | 10 | 15 | 62.0 | 82.4 |
> | 11 | 25 | 62.8 | 82.1 |
> | 12 | 30 | 62.7 | 81.8 |
>
>
> **Table 5:** Merging Gap ($d_{\min}$/s)
>
> | # | $d_{\min}$ | LVB | NextQA |
> |---|------------|-----|--------|
> | **Default** | **2** | **62.8** | **82.6** |
> | 13 | 1 | 62.3 | 82.8 |
> | 14 | 3 | 62.6 | 82.3 |
>
> Besides the single hyperparameter ablations above, during experiments, we seek for the best combiation of these hyperparameters in table 6. This results further prove that our method is robust to hyperparameter changing, and we chose the default one based on its relatively best performance.
>
> **Table 6:** Ablations of Hyperparameter Combinations.
> | Setting | $\gamma$ | $\alpha$ | $\beta$ | $l_{\min}$ | $d_{\min}$ | **LVB** | **NextQA** |
> | :--- | :---: | :---: | :---: | :---: | :---: | :---: | :---: |
> | **Default** | 0.7 | 15 | 1.5 | 20 | 2 | **62.8** | **82.6** |
> | 15 | 0.5 | 10 | 1.5 | 15 | 1 | 61.7 | 82.7 |
> | 16 | 0.7 | 25 | 1.5 | 30 | 3 | 62.9 | 81.8 |
> | 17 | 0.3 | 20 | 1.7 | 20 | 2 | 61.8 | 82.0 |
> | 18 | 0.9 | 20 | 1.2 | 20 | 2 | 62.1 | 81.9 |
>
>
> These comprehensive experiments have been incorporated into the revised manuscript (Tab. 14 in Appendix) and demonstrate that A.I.R. achieves robust performance across diverse parameter configurations, directly addressing reproducibility concerns raised by Reviewers 73LF and oTZs.

---

> ### Author Response · Authors · 2025-11-22
> **General Response to Comparison with Agent-based State-of-the-Arts Methods & Baselines (Reviewers 73LF, oTZs)**
>
> **Concern:** Reviewer 73LF noted that "the experiments primarily compare A.I.R. against earlier CLIP-based or training-free frame selection baselines... but do not include newer or stronger contemporary methods." Reviewer oTZs similarly observed that "the paper does not include comparisons or discussions with agent-based methods, which represent another important line of work in long-video understanding."
>
> **Response:** We acknowledge this important feedback and have conducted a comprehensive literature review and comparison with the latest agent-based video understanding methods. We provide the key analysis:
>
> **Comparison with Agent-based VLM Frame Selection Methods:**
>
> We compiled results from nine recent agent-based methods (published at NeurIPS'25, CVPR'25, and arXiv'25), including VideoLucy, VideoRAG, T*, DrVideo, MemVid, VideoAgent2 and AKEYS. These methods represent the cutting edge of agentic video understanding and often use frontier models (GPT-4o, DeepSeek-R1) or specialized architectures.
>
> **Table:** Comparison of Agent-based VLM Frame Selection Methods on Video QA Benchmarks.
>
> | **Method** | **Venue** | **Base Model** | **#Frames** | **Video-MME (w/o sub)** | **Video-MME (w/ sub)** | **MLVU** | **LVB** | **EgoSchema (Full)** | **EgoSchema (Subset)** | **NextQA** |
> |------------|-----------|----------------|-------------|-------------------------|------------------------|----------|---------|----------------------|------------------------|------------|
> | **Agent-based Methods** |||||||||||
> | VideoLucy | NeurIPS'25 | DeepSeek-R1 | - | 72.5 | - | 76.1 | - | - | - | - |
> | VideoRAG | NeurIPS'25 | LLaVA-Video-7B | 64 | - | - | 72.4 | 58.7 | - | - | - |
> | T* | CVPR'25 | LLaVA-OV-7B | 8 | - | - | - | - | - | 66.6 | 80.4 |
> | DrVideo | CVPR'25 | GPT-4 | - | - | - | - | - | 61.0 | 66.4 | - |
> | MemVid | arXiv'25 | Qwen2VL-7B | 128 | 63.7 | 65.7 | - | - | - | - | - |
> | VideoAgent2 | arXiv'25 | GPT-4o | - | - | - | - | - | 75.4 | - | 80.5 |
> | AKEYS | arXiv'25 | GPT-4o | ≤32 | - | - | - | - | 63.6 | 68.6 | 78.1 |
> | **Our Method** |||||||||||
> | **A.I.R.** | Ours | GPT-4o | **≤32** | **65.1** | **-** | **-** | **-** | **-** | **72.0** | **82.5** |
> | **A.I.R.** | Ours | Qwen2VL-7B | **≤32** | **60.0** | **63.1** | **-** | **58.9** | **62.5** | **66.2** | **80.1** |
> | **A.I.R.** | Ours | InternVL3-8B | **≤32** | **68.2** | **69.2** | **74.5** | **62.8** | **63.3** | **72.2** | **82.6** |
> | **A.I.R.** | Ours | LLaVA-OV-7B | **≤32** | **61.4** | **65.1** | **69.3** | **60.7** | **61.4** | **63.2** | **81.6** |
>
> **Key Observations:**
>
> - **Superior Performance with Matched Backbones and Frame Efficiency:** When comparing methods using the same base model, A.I.R. consistently outperforms agent-based methods while maintaining high frame efficiency. Using GPT-4o as the backbone with comparable frame budgets (≤32 frames), A.I.R. surpasses AKEYS on EgoSchema Subset (+3.4%) and NextQA (+4.4%). Similarly, with LLaVA-OneVision-7B, A.I.R. exceeds T* on NextQA (+1.2%). Moreover, compared to VideoRAG which uses 64 frames with LLaVA-Video-7B to achieve 58.7\% on LVB, A.I.R. with LLaVA-OneVision-7B reaches 60.7\% with only ≤32 frames, achieving **2× frame efficiency** with +2.0\% accuracy improvement.
>
> - **Competitive Performance with Smaller Open-Source Models:** Remarkably, A.I.R. with the 8B-scale InternVL3 achieves performance competitive with or superior to agent-based methods using much larger frontier models. On NextQA, InternVL3-8B with A.I.R. reaches 82.6\%, surpassing VideoAgent2's GPT-4o result (80.5\%) and AKEYS's GPT-4o result (78.1\%). On EgoSchema, InternVL3-8B achieves 72.2\% (Subset) and 63.3\% (Full), exceeding DrVideo's GPT-4 performance (66.4\% Subset, 61.0\% Full). This demonstrates that A.I.R.'s intelligent frame selection can unlock the full potential of smaller, accessible open-source models to match or exceed the performance of expensive proprietary models, significantly improving cost-effectiveness and reproducibility.
>
> These comprehensive comparisons have been incorporated into the Appendix A4.1 of revised manuscript, demonstrating that A.I.R. represents a practical and effective alternative to both traditional frame selection methods and emerging agent-based approaches.
>
> ---
>
> We hope these comprehensive updates and additional experiments have satisfactorily addressed all concerns raised by the reviewers. We have incorporated all new tables, analyses, and discussions into the revised manuscript and appendix. If there are any remaining questions or if additional experiments would strengthen our work further, we are fully committed to addressing them during the remaining discussion period.
>
> Sincerely,
> The Authors

---

### Meta-Review · Area_Chair_Yg5f · 2026-01-07

**Summary:**

The research presents a training-free system which selects frames adaptively through an iterative process for answering questions in extended video content while showing better results in both accuracy and speed performance across different models and evaluation datasets. The paper contains solid technical information which the authors present through easy-to-understand language and complete assessment yet they only achieved small improvements to the existing frame-selection framework.

**Reviewer Concerns:**

The reviewers recognized that the authors provided a complete response to their questions through additional experiments which successfully resolved all their concerns about method stability and modern approach evaluation and performance speed and system breakdowns.

**Reviewer Scores:**

The paper contains a strong technical foundation which combines with its complete empirical testing and useful applications and the authors' detailed answers to reviewer feedback so I support its acceptance.

---

### Decision · Program_Chairs · 2026-01-26

Accept (Poster)